# ADVANTAGE-WEIGHTED REGRESSION: SIMPLE AND SCALABLE OFF-POLICY REINFORCEMENT LEARNING

## ABSTRACT

In this paper, we aim to develop a simple and scalable reinforcement learning algorithm that uses standard supervised learning methods as subroutines. Our goal is an algorithm that utilizes only simple and convergent maximum likelihood loss functions, while also being able to leverage off-policy data. Our proposed approach, which we refer to as *advantage-weighted regression* (AWR), consists of two standard supervised learning steps: one to regress onto target values for a value function, and another to regress onto weighted target actions for the policy. The method is simple and general, can accommodate continuous and discrete actions, and can be implemented in just a few lines of code on top of standard supervised learning methods. We provide a theoretical motivation for AWR and analyze its properties when incorporating off-policy data from experience replay. We evaluate AWR on a suite of standard OpenAI Gym benchmark tasks, and show that it achieves competitive performance compared to a number of well-established state-of-the-art RL algorithms. AWR is also able to acquire more effective policies than most off-policy algorithms when learning from purely static datasets with no additional environmental interactions. Furthermore, we demonstrate our algorithm on challenging continuous control tasks with highly complex simulated characters. (Video[1])

## 1 INTRODUCTION

Model-free reinforcement learning can be a general and effective methodology for training agents to acquire sophisticated behaviors with minimal assumptions on the underlying task (Mnih et al., 2015; Heess et al., 2017; Pathak et al., 2017). However, reinforcement learning algorithms can be substantially more complex to implement and tune than standard supervised learning methods. Arguably the simplest reinforcement learning methods are policy gradient algorithms (Sutton et al., 2000), which directly differentiate the expected return and perform gradient ascent. Unfortunately, these methods can be notoriously unstable and are typically on-policy (or nearly on-policy), often requiring a substantial number of samples to learn effective behaviors. Our goal is to develop a reinforcement learning algorithm that is simple, easy to implement, and can readily incorporate off-policy experience data.

In this work, we propose advantage-weighted regression (AWR), a simple off-policy algorithm for model-free RL. Each iteration of the AWR algorithm simply consists of two supervised regression steps: one for training a value function baseline via regression onto cumulative rewards, and another for training the policy via weighted regression. The complete algorithm is shown in Algorithm 1.

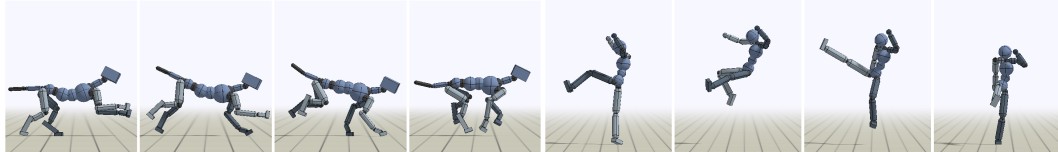

Figure 1: Complex simulated character trained using advantage-weighted regression. **Left:** Humanoid performing a spinkick. **Right:** Dog performing a canter.

---

[1]Supplementary video: sites.google.com/view/awr-supp/

AWR can accommodate continuous and discrete actions, and can be implemented in just a few lines of code on top of standard supervised learning methods. Despite its simplicity, we find that AWR achieves competitive results when compared to commonly used on-policy and off-policy RL algorithms, and can effectively incorporate fully off-policy data, which has been a challenge for other RL algorithms. Our derivation presents an interpretation of AWR as a constrained policy optimization procedure, and provides a theoretical analysis of the use of off-policy data with experience replay.

We first revisit the original formulation of reward-weighted regression, an on-policy RL method that utilizes supervised learning to perform policy updates, and then propose a number of new design decisions that significantly improve performance on a suite of standard continuous control benchmark tasks. We then provide a theoretical analysis of AWR, including the capability to incorporate off-policy data with experience replay. Although the design of AWR involves only a few simple design decisions, we show experimentally that these additions provide for a large improvement over previous methods for regression-based policy search, such as reward-weighted regression (RWR) (Peters & Schaal, 2007), while also being substantially simpler than more modern methods, such as MPO (Abdolmaleki et al., 2018). We show that AWR achieves competitive performance when compared to several well-established state-of-the-art on-policy and off-policy algorithms. We further demonstrate our algorithm on challenging control tasks with complex simulated characters.

## 2 PRELIMINARIES

In reinforcement learning, the objective is to learn a control policy that enables an agent to maximize its expected return for a given task. At each time step $t$, the agent observes the state of the environment $\mathbf{s}_t \in \mathcal{S}$, and samples an action $\mathbf{a}_t \in \mathcal{A}$ from a policy $\mathbf{a}_t \sim \pi(\mathbf{a}_t|\mathbf{s}_t)$. The agent then applies that action, which results in a new state $\mathbf{s}_{t+1}$ and a scalar reward $r_t = r(\mathbf{s}_t, \mathbf{a}_t)$. The goal is to learn an optimal policy that maximizes the agent's expected discounted return $J(\pi)$,

$$J(\pi) = \mathbb{E}_{\tau \sim p_\pi(\tau)} \left[ \sum_{t=0}^{\infty} \gamma^t r_t \right] = \mathbb{E}_{\mathbf{s} \sim d_\pi(\mathbf{s})} \mathbb{E}_{a \sim \pi(\mathbf{a}|\mathbf{s})} \left[ r(\mathbf{s}, \mathbf{a}) \right], \tag{1}$$

where $p_\pi(\tau)$ represents the likelihood of a trajectory $\tau = \{(\mathbf{s}_0, \mathbf{a}_0, r_0), (\mathbf{s}_1, \mathbf{a}_1, r_1), ...\}$ under a policy $\pi$, and $\gamma \in [0, 1)$ is the discount factor. $d_\pi(\mathbf{s}) = \sum_{t=0}^{\infty} \gamma^t p(\mathbf{s}_t = \mathbf{s}|\pi)$ represents the unnormalized discounted state distribution induced by the policy $\pi$ (Sutton & Barto, 1998), and $p(\mathbf{s}_t = \mathbf{s}|\pi)$ is the likelihood of the agent being in state $\mathbf{s}$ after following $\pi$ for $t$ timesteps. A popular class of algorithms for solving this problem is policy gradient (PG) methods, which directly estimates the gradient of the expected return with respect to the policy parameters $\nabla_\pi J(\pi)$, and then updates the policy with gradient ascent. Basic PG algorithms are generally on-policy methods, which require the data to be sampled from the same policy that is being optimized. This can result in poor sample efficiency, but PG algorithms can be modified to utilize off-policy data.

An alternative class of RL methods is expectation-maximization algorithms. Instead of estimating the gradient of the expected return, EM algorithms first construct an estimate of the optimal policy using a dataset of experiences (E-step), and then projects this estimate onto the space of parameterized policies (M-step). An early example of an EM-based RL algorithm is reward-weighted regression (RWR) (Peters et al., 2010). At each iteration, the E-step constructs an estimate of the optimal policy according to $\pi^*(\mathbf{a}|\mathbf{s}) \propto \pi_k(\mathbf{a}|\mathbf{s})\exp(\mathcal{R}_{\mathbf{s},\mathbf{a}}/\beta)$, where $\pi_k$ represents the policy at the $k$th iteration of the algorithm, $\mathcal{R}_{\mathbf{s},\mathbf{a}} = \sum_{t=0}^{\infty} \gamma^t r_t$ is the return, and $\beta > 0$ is a temperature parameter. Then the M-step projects $\pi^*$ onto the space of parameterized policies by solving a supervised regression problem:

$$\pi_{k+1} = \arg\max_\pi \quad \mathbb{E}_{\mathbf{s} \sim d_{\pi_k}(\mathbf{s})} \mathbb{E}_{\mathbf{a} \sim \pi_k(\mathbf{a}|\mathbf{s})} \left[ \log \pi(\mathbf{a}|\mathbf{s}) \exp\left( \frac{1}{\beta} \mathcal{R}_{\mathbf{s},\mathbf{a}} \right) \right]. \tag{2}$$

The RWR update can be interpreted as fitting a new policy $\pi_{k+1}$ to samples collected under the current policy $\pi_k$, where the likelihood of each action is weighted by the exponentiated return received for that action. Since EM algorithms do not directly estimate the gradient of the expected return with respect to the current policy, they can be more amenable to learning from off-policy data.

---

**Algorithm 1** Advantage-Weighted Regression

---

1: $\pi_1 \leftarrow$ random policy
2: $\mathcal{D} \leftarrow \emptyset$
3: **for** iteration $k = 1, ..., k_{\max}$ **do**
4:     add trajectories $\{\tau_i\}$ sampled via $\pi_k$ to $\mathcal{D}$
5:     $V_k^{\mathcal{D}} \leftarrow \arg\min_V \; \mathbb{E}_{\mathbf{s},\mathbf{a} \sim \mathcal{D}} \left[ \left|\left| \mathcal{R}_{\mathbf{s},\mathbf{a}}^{\mathcal{D}} - V(\mathbf{s}) \right|\right|^2 \right]$
6:     $\pi_{k+1} \leftarrow \arg\max_\pi \; \mathbb{E}_{\mathbf{s},\mathbf{a} \sim \mathcal{D}} \left[ \log \pi(\mathbf{a}|\mathbf{s}) \exp\left( \frac{1}{\beta} \left( \mathcal{R}_{\mathbf{s},\mathbf{a}}^{\mathcal{D}} - V_k^{\mathcal{D}}(\mathbf{s}) \right) \right) \right]$
7: **end for**

---

## 3   ADVANTAGE-WEIGHTED REGRESSION

In this work, we present advantage-weighted regression (AWR), a simple off-policy RL algorithm based on reward-weighted regression. We first provide an overview of the complete advantage-weighted regression algorithm, and then describe its theoretical motivation and analyze its properties. The complete AWR algorithm is summarized in Algorithm 1. Each iteration $k$ of AWR consists of the following simple steps. First, the current policy $\pi_k(\mathbf{a}|\mathbf{s})$ is used to sample a batch of trajectories $\{\tau_i\}$ that are then stored in the replay buffer $\mathcal{D}$, which is structured as a first-in first-out (FIFO) queue, as is common for off-policy reinforcement learning algorithms (Mnih et al., 2015; Lillicrap et al., 2016). Then, the entire buffer $\mathcal{D}$ is used to fit a value function $V_k^{\mathcal{D}}(\mathbf{s})$ to the trajectories in the replay buffer, which can be done with simple Monte Carlo return estimates $\mathcal{R}_{\mathbf{s},\mathbf{a}}^{\mathcal{D}} = \sum_{t=0}^{T} \gamma^t r_t$. Finally, the same buffer is used to fit a new policy using *advantage-weighted* regression, where each state-action pair in the buffer is weighted according to the exponentiated advantage $\exp(\frac{1}{\beta} A^{\mathcal{D}}(\mathbf{s}, \mathbf{a}))$, with the advantage given by $A^{\mathcal{D}}(\mathbf{s}, \mathbf{a}) = \mathcal{R}_{\mathbf{s},\mathbf{a}}^{\mathcal{D}} - V^{\mathcal{D}}(\mathbf{s})$ and $\beta$ is a hyperparameter. AWR uses only supervised regression as learning subroutines, making the algorithm very simple to implement. In the following subsections, we first motivate the algorithm as an approximation to a constrained policy search problem, and then extend our analysis to incorporate experience replay.

### 3.1   DERIVATION

In this section, we derive the AWR algorithm as an approximate optimization of a constrained policy search problem. Our goal is to find a policy that maximizes the expected *improvement* $\eta(\pi) = J(\pi) - J(\mu)$ over a sampling policy $\mu(\mathbf{a}|\mathbf{s})$. We first derive AWR for the setting where the sampling policy is a single Markovian policy. Then, in the next section, we extend our result to the setting where the data is collected from multiple policies, as in the case of experience replay that we use in practice. The expected improvement $\eta(\pi)$ can be expressed in terms of the advantage $A^\mu(\mathbf{s}, \mathbf{a}) = \mathcal{R}_{\mathbf{s},\mathbf{a}}^\mu - V^\mu(\mathbf{s})$ with respect to the sampling policy $\mu$ (Kakade & Langford, 2002; Schulman et al., 2015):

$$\eta(\pi) = \mathbb{E}_{\mathbf{s} \sim d_\pi(\mathbf{s})} \mathbb{E}_{\mathbf{a} \sim \pi(\mathbf{a}|\mathbf{s})} \left[ A^\mu(\mathbf{s}, \mathbf{a}) \right] = \mathbb{E}_{\mathbf{s} \sim d_\pi(\mathbf{s})} \mathbb{E}_{\mathbf{a} \sim \pi(\mathbf{a}|\mathbf{s})} \left[ \mathcal{R}_{\mathbf{s},\mathbf{a}}^\mu - V^\mu(\mathbf{s}) \right], \quad (3)$$

where $\mathcal{R}_{\mathbf{s},\mathbf{a}}^\mu$ denotes the return obtained by performing action $\mathbf{a}$ in state $\mathbf{s}$ and following $\mu$ for the following timesteps, and $V^\mu(\mathbf{s}) = \int_a \mu(\mathbf{a}|\mathbf{s}) \mathcal{R}_{\mathbf{s}}^{\mathbf{a}} \, d\mathbf{a}$ corresponds to the value function of $\mu$. This objective differs from the ones used in the derivations of related algorithms, such as RWR and REPS (Peters & Schaal, 2007; Peters et al., 2010; Abdolmaleki et al., 2018), which maximize the expected return $J(\pi)$ instead of the expected improvement. The expected improvement directly gives rise to an objective that involves the advantage. We will see later that this yields weights for the policy update that differ in a subtle but important way from standard reward-weighted regression. As we show in our experiments, this difference results in a large empirical improvement.

The objective in Equation 3 can be difficult to optimize due to the dependency between $d_\pi(\mathbf{s})$ and $\pi$, as well as the need to collect samples from $\pi$. Following Schulman et al. (2015), we can instead optimize an approximation $\hat{\eta}(\pi)$ of $\eta(\pi)$ using the state distribution of $\mu$:

$$\hat{\eta}(\pi) = \mathbb{E}_{\mathbf{s} \sim d_\mu(\mathbf{s})} \mathbb{E}_{\mathbf{a} \sim \pi(\mathbf{a}|\mathbf{s})} \left[ \mathcal{R}_{\mathbf{s},\mathbf{a}}^\mu - V^\mu(\mathbf{s}) \right]. \quad (4)$$

Here, $\hat{\eta}(\pi)$ matches $\eta(\pi)$ to first order (Kakade & Langford, 2002), and provides a good estimate of $\eta$ if $\pi$ and $\mu$ are close in terms of the KL-divergence (Schulman et al., 2015). Using this objective,

we can formulate the following *constrained* policy search problem:

$$\arg\max_{\pi} \quad \int_{\mathbf{s}} d_{\mu}(\mathbf{s}) \int_{\mathbf{a}} \pi(\mathbf{a}|\mathbf{s}) \left[ \mathcal{R}_{\mathbf{s},\mathbf{a}}^{\mu} - V^{\mu}(\mathbf{s}) \right] \, d\mathbf{a} \, d\mathbf{s} \tag{5}$$

$$\text{s.t.} \quad \int_{\mathbf{s}} d_{\mu}(\mathbf{s}) \mathrm{D}_{\mathrm{KL}} \left( \pi(\cdot|\mathbf{s}) || \mu(\cdot|\mathbf{s}) \right) d\mathbf{s} \leq \epsilon. \tag{6}$$

The constraint in Equation 6 ensures that the new policy $\pi$ is close to the data distribution of $\mu$, and therefore the surrogate objective $\hat{\eta}(\pi)$ remains a reasonable approximation to $\eta(\pi)$. We refer the reader to Schulman et al. (2015) for a detailed derivation and an error bound.

We can derive AWR as an approximate solution to this constrained optimization. This derivation follows a similar procedure as Peters et al. (2010), and begins by forming the Langrangian of the constrained optimization problem presented above,

$$\mathcal{L}(\pi, \beta) = \int_{\mathbf{s}} d_{\mu}(\mathbf{s}) \int_{\mathbf{a}} \pi(\mathbf{a}|\mathbf{s}) \left[ \mathcal{R}_{\mathbf{s},\mathbf{a}}^{\mu} - V^{\mu}(\mathbf{s}) \right] \, d\mathbf{a} \, d\mathbf{s} + \beta \left( \epsilon - \int_{\mathbf{s}} d_{\mu}(\mathbf{s}) \mathrm{D}_{\mathrm{KL}} \left( \pi(\cdot|\mathbf{s}) || \mu(\cdot|\mathbf{s}) \right) d\mathbf{s} \right), \tag{7}$$

where $\beta$ is a Lagrange multiplier. Differentiating $\mathcal{L}(\pi, \beta)$ with respect to $\pi(\mathbf{a}|\mathbf{s})$ and solving for the optimal policy $\pi^*$ results in the following expression for the optimal policy

$$\pi^*(\mathbf{a}|\mathbf{s}) = \frac{1}{Z(\mathbf{s})} \, \mu(\mathbf{a}|\mathbf{s}) \exp\left( \frac{1}{\beta} \left( \mathcal{R}_{\mathbf{s},\mathbf{a}}^{\mu} - V^{\mu}(\mathbf{s}) \right) \right), \tag{8}$$

with $Z(\mathbf{s})$ being the partition function. A detailed derivation is available in Appendix A. If $\pi$ is represented by a function approximator (e.g., a neural network), a new policy can be obtained by projecting $\pi^*$ onto the manifold of parameterized policies,

$$\arg\min_{\pi} \quad \mathbb{E}_{\mathbf{s} \sim \mathcal{D}} \left[ \mathrm{D}_{\mathrm{KL}} \left( \pi^*(\cdot|\mathbf{s}) || \pi(\cdot|\mathbf{s}) \right) \right] \tag{9}$$

$$= \arg\max_{\pi} \quad \mathbb{E}_{\mathbf{s} \sim d_{\mu}(\mathbf{s})} \mathbb{E}_{\mathbf{a} \sim \mu(\mathbf{a}|\mathbf{s})} \left[ \log \pi(\mathbf{a}|\mathbf{s}) \exp\left( \frac{1}{\beta} \left( \mathcal{R}_{\mathbf{s},\mathbf{a}}^{\mu} - V^{\mu}(\mathbf{s}) \right) \right) \right]. \tag{10}$$

While this derivation for AWR largely follows the derivations used in prior work (Peters et al., 2010; Abdolmaleki et al., 2018), our expected improvement objective introduces a baseline $V^{\mu}(\mathbf{s})$ to the policy update, which as we show in our experiments, is a crucial component for an effective algorithm. A similar advantage-weighting scheme has been previously used for fitted Q-iteration (Neumann & Peters, 2009), where the policy is given by $\pi(\mathbf{a}|\mathbf{s}) = \frac{1}{Z(\mathbf{s})} \exp\left( \left( \mathcal{R}_{\mathbf{s},\mathbf{a}}^{\mu} - V^{\mu}(\mathbf{s}) \right) / \beta \right)$. In this definition, the likelihood of an action does not depend on its likelihood under the sampling distribution, and therefore does not enforce a trust region with respect to the sampling distribution. Next, we extend AWR to incorporate experience replay for off-policy training, where the sampling policy is no longer a single policy, but rather a mixture of policies from past iterations.

### 3.2 EXPERIENCE REPLAY AND OFF-POLICY LEARNING

A crucial design decision of AWR is the choice of sampling policy $\mu(\mathbf{a}|\mathbf{s})$. Standard implementations of RWR typically follow an on-policy approach, where the sampling policy is selected to be the current policy $\mu(\mathbf{a}|\mathbf{s}) = \pi_k(\mathbf{a}|\mathbf{s})$ at iteration $k$. This can be sample inefficient, as data collected at each iteration of the algorithms are discarded after a single update iteration. Importance sampling can be incorporated into RWR to reuse data from previous iterations, but at the cost of larger variance from the importance sampling estimator (Kober & Peters, 2009). Instead, we can improve sample efficiency of AWR by incorporating experience replay and explicitly accounting for training data from a mixture of multiple prior policies. As described in Algorithm 1, at each iteration, AWR collects a batch of data using the latest policy $\pi_k$, and then stores this data in a replay buffer $\mathcal{D}$, which also contains data collected from previous policies $\{\pi_1, \cdots, \pi_k\}$. The value function and policy are then updated using samples drawn from $\mathcal{D}$. This replay strategy is analogous to modeling the sampling policy as a mixture of policies from previous iterations $\mu_k(\tau) = \sum_{i=1}^{k} w_i \pi_i(\tau)$, where $\pi_i(\tau) = p(\tau|\pi_i)$ represents the likelihood of a trajectory $\tau$ under a policy $\pi_i$ from the $i$th iteration, and the weights $\sum_i w_i = 1$ specify the probabilities of selecting each policy $\pi_i$.

We now extend the derivation from the previous section to the off-policy setting with experience replay, and show that Algorithm 1 indeed optimizes the expected improvement over a sampling policy

modeled by the replay buffer. Given a replay buffer consisting of trajectories from past policies, the joint state-action distribution of $\mu$ is given by $\mu(\mathbf{s}, \mathbf{a}) = \sum_{i=1}^{k} w_i d_{\pi_i}(\mathbf{s}) \pi_i(\mathbf{a}|\mathbf{s})$, and similarly for the marginal state distribution $d_\mu(\mathbf{s}) = \sum_{i=1}^{k} w_i d_{\pi_i}(\mathbf{s})$. The *expected improvement* can now be expressed with respect to the set of sampling policies in the replay buffer: $\eta(\pi) = J(\pi) - \sum_i w_i J(\pi_i)$. Similar to Equation 3, $\eta(\pi)$ can be expressed in terms of the advantage $A^{\pi_i}(\mathbf{s}, \mathbf{a}) = \mathcal{R}_{\mathbf{s},\mathbf{a}}^{\pi_i} - V^{\pi_i}(\mathbf{s})$ of each sampling policies,

$$\eta(\pi) = J(\pi) - \sum_i w_i J(\pi_i) = \mathbb{E}_{\mathbf{s} \sim d_\pi(\mathbf{s})} \mathbb{E}_{\mathbf{a} \sim \pi(\mathbf{a}|\mathbf{s})} \left[ \sum_i w_i A^{\pi_i}(\mathbf{s}, \mathbf{a}) \right]. \tag{11}$$

As before, we can optimize an approximation $\hat{\eta}(\pi)$ of $\eta(\pi)$ using the state distribution of $\mu$,

$$\hat{\eta}(\pi) = \mathbb{E}_{\mathbf{s} \sim d_\mu(\mathbf{s})} \mathbb{E}_{\mathbf{a} \sim \pi(\mathbf{a}|\mathbf{s})} \left[ A^{\pi_i}(\mathbf{s}, \mathbf{a}) \right] = \sum_{i=1}^{k} w_i \left( \mathbb{E}_{\mathbf{s} \sim d_{\pi_i}(\mathbf{s})} \mathbb{E}_{\mathbf{a} \sim \pi(\mathbf{a}|\mathbf{s})} \left[ A^{\pi_i}(\mathbf{s}, \mathbf{a}) \right] \right) \tag{12}$$

In Appendix B, we show that the update procedure in Algorithm 1 optimizes the following objective:

$$\arg\max_\pi \quad \sum_{i=1}^{k} w_i \left( \mathbb{E}_{\mathbf{s} \sim d_{\pi_i}(\mathbf{s})} \mathbb{E}_{\mathbf{a} \sim \pi(\mathbf{a}|\mathbf{s})} \left[ A^{\pi_i}(\mathbf{s}, \mathbf{a}) \right] \right) \tag{13}$$

$$\text{s.t.} \quad \mathbb{E}_{\mathbf{s} \sim d_\mu(\mathbf{s})} \left[ D_{\text{KL}} \left( \pi(\cdot|\mathbf{s}) || \mu(\cdot|\mathbf{s}) \right) \right] \leq \epsilon, \tag{14}$$

where $\mu(\mathbf{a}|\mathbf{s}) = \frac{\mu(\mathbf{s},\mathbf{a})}{d_\mu(\mathbf{s})} = \frac{\sum_i w_i d_{\pi_i}(\mathbf{s}) \pi_i(\mathbf{a}|\mathbf{s})}{\sum_j w_j d_{\pi_j}(\mathbf{s})}$ represents the conditional action distribution defined by the replay buffer. This objective can be solved via the Lagrangian to yield the following update:

$$\arg\max_\pi \quad \sum_{i=1}^{k} w_i \, \mathbb{E}_{\mathbf{s} \sim d_{\pi_i}(\mathbf{s})} \mathbb{E}_{\mathbf{a} \sim \pi_i(\mathbf{a}|\mathbf{s})} \left[ \log \pi(\mathbf{a}|\mathbf{s}) \exp \left( \frac{1}{\beta} \left( \mathcal{R}_{\mathbf{s},\mathbf{a}}^{\pi_i} - \frac{\sum_j w_j d_{\pi_j}(\mathbf{s}) V^{\pi_j}(\mathbf{s})}{\sum_j w_j d_{\pi_j}(\mathbf{s})} \right) \right) \right], \tag{15}$$

where the expectations can be approximated by simply sampling from $\mathcal{D}$ following Line 6 of Algorithm 1. A detailed derivation is available in Appendix B. Note, the baseline in the exponent now consists of an average of the value functions of the different policies. One approach for estimating this quantity would be to fit separate value functions $V^{\pi_i}$ for each policy. However, if only a small amount of data is available from each policy, then $V^{\pi_i}$ could be highly inaccurate (Fu et al., 2019). Therefore, instead of learning separate value functions, we fit a single *mean* value function $\bar{V}(\mathbf{s})$ that directly estimates the weighted average of $V^{\pi_i}$'s,

$$\bar{V} = \arg\min_V \sum_i w_i \, \mathbb{E}_{\mathbf{s}, \sim d_{\pi_i}(\mathbf{s})} \mathbb{E}_{\mathbf{a} \sim \pi_i(\mathbf{a}|\mathbf{s})} \left[ ||\mathcal{R}_{\mathbf{s},\mathbf{a}}^{\pi_i} - V(\mathbf{s})||^2 \right] \tag{16}$$

This loss can also be approximated by simply sampling from the replay buffer following Line 5 of Algorithm 1. The optimal solution $\bar{V}(\mathbf{s}) = \frac{\sum_i w_i d_{\pi_i}(\mathbf{s}) V^{\pi_i}(\mathbf{s})}{\sum_j w_j d_{\pi_j}(\mathbf{s})}$ is exactly the baseline in Equation 15.

## 3.3 IMPLEMENTATION DETAILS

Finally, we discuss several design decisions that are important for a practical implementation of AWR. An overview of AWR is provided in Algorithm 1. The policy update in Equation 10 requires sampling states from the *discounted* state distribution $d_\mu(\mathbf{s})$. However, we found that simply sampling states uniformly from $\mathcal{D}$ was also effective, and simpler to implement. This is a common strategy used in standard implementations of RL algorithms (Dhariwal et al., 2017). When updating the value function and policy, Monte Carlo estimates can be used to approximate the expected return $\mathcal{R}_{\mathbf{s},\mathbf{a}}^{\mathcal{D}}$ of samples in $\mathcal{D}$, but this can result in a high-variance estimate. Instead, we opt to approximate $\mathcal{R}_{\mathbf{s},\mathbf{a}}^{\mathcal{D}}$ using TD($\lambda$) to obtain a lower-variance estimate (Sutton & Barto, 1998). TD($\lambda$) is applied by bootstrapping with the value function $V_{k-1}^{\mathcal{D}}(\mathbf{s})$ from the previous iteration. A simple Monte Carlo return estimator can also be used though, as shown in our experiments, but this produces somewhat worse results. To further simplify the algorithm, instead of adaptively updating the Lagrange multiplier $\beta$, as is done in previous methods (Peters & Schaal, 2007; Peters et al., 2010; Abdolmaleki et al., 2018), we find that simply using a fixed constant for $\beta$ is also effective. The weights $\omega_{\mathbf{s},\mathbf{a}}^{\mathcal{D}} = \exp \left( \frac{1}{\beta} \left( \mathcal{R}_{\mathbf{s},\mathbf{a}}^{\mathcal{D}} - V^{\mathcal{D}}(\mathbf{s}) \right) \right)$ used to update the policy can occasionally assume excessively large values, which can cause gradients to explode. We therefore apply weight clipping $\hat{\omega}_{\mathbf{s},\mathbf{a}}^{\mathcal{D}} = \min \left( \omega_{\mathbf{s},\mathbf{a}}^{\mathcal{D}}, \, \omega_{\max} \right)$ with a threshold $\omega_{\max}$ to mitigate issues due to exploding weights.

## 4 RELATED WORK

Existing RL methods can be broadly categorized into on-policy and off-policy algorithms (Sutton & Barto, 1998). On-policy algorithms generally update the policy using data collected from the same policy. A popular class of on-policy algorithms is policy gradient methods (Williams, 1992; Sutton et al., 2000), which have been shown to be effective for a diverse array of complex tasks (Heess et al., 2017; Pathak et al., 2017; Peng et al., 2018; Rajeswaran et al., 2018). However, on-policy algorithms are typically data inefficient, requiring a large number of interactions with the environment. Off-policy algorithms improve sample efficiency by enabling a policy to be trained using data from other sources, such as data collected from different agents or data from previous iterations of the algorithm. Importance sampling is a simple strategy for incorporating off-policy data (Sutton & Barto, 1998; Meuleau et al., 2000; Hachiya et al., 2009), but can introduce optimization instabilities due to the potentially large variance of the importance sampling estimator. Dynamic programming methods based on Q-function learning can also leverage off-policy data (Precup et al., 2001; Mnih et al., 2015; Lillicrap et al., 2016; Gu et al., 2016; Haarnoja et al., 2018b). But these methods can be notoriously unstable, and in practice, require a variety of stabilization techniques to ensure more consistent performance (Hasselt et al., 2016; Wang et al., 2016; Munos et al., 2016; Hessel et al., 2017; Fujimoto et al., 2018; Nachum et al., 2018; Fu et al., 2019). Furthermore, it can be difficult to apply these methods to learn from fully off-policy data, where an agent is unable to collect additional environmental interactions (Fujimoto et al., 2019; Kumar et al., 2019).

Alternatively, policy search can also be formulated under an expectation-maximization framework. This approach has lead to a variety of EM-based RL algorithms (Peters et al., 2010; Neumann, 2011; Abdolmaleki et al., 2018), an early example of which is reward-weighted regression (RWR) (Peters & Schaal, 2007). RWR presents a simple on-policy RL algorithm that casts policy search as a supervised regression problem. A similar algorithm, relative entropy policy search (REPS) (Peters et al., 2010), can also be derived from the dual formulation of a constrained policy search problem. RWR has a number appealing properties: it has a very simple update rule, and since each iteration corresponds to supervised learning, it can be more stable and easier to implement than many of the previously mentioned RL methods. Despite these advantages, RWR has not been shown to be an effective RL algorithm when combined with neural network function approximators, as demonstrated in prior work and our own experiments (Schulman et al., 2015; Duan et al., 2016). In this work, we propose a number of modifications to the formulation of RWR to produce an effective off-policy deep RL algorithm, while still retaining much of the simplicity of previous methods.

Policy updates using supervised regression have been used in a number of prior work. The optimization problem being solved in REPS is similar to AWR (Peters et al., 2010), but REPS optimizes the expected return instead of the expected improvement. The weights in REPS also contains a Bellman error term that superficially resembles advantages, but are computed using a linear value function derived from a feature matching constraint. Learning the REPS value function requires minimization of a dual function, which is a complex function of the Bellman error, while the value function in AWR can be learned with simple supervised regression. More recently, Abdolmaleki et al. (2018) proposed MPO, a deep RL variant of REPS, which applies a partial EM algorithm for policy optimization. The method first fits a Q-function of the current policy via bootstrapping, and then performs a policy improvement step with respect to this Q-function under a trust region constraint that penalizes large policy changes. MPO uses off-policy data for training a Q-function critic via bootstrapping and employs Retrace($\lambda$) for off-policy correction (Munos et al., 2016). In contrast, AWR is substantially simpler, as it can simply fit a value function to the observed returns in a replay buffer, and performs weighted supervised regression on the actions to fit the policy. Oh et al. (2018) proposed self-imitation learning (SIL), which augments policy gradient algorithms with an auxiliary behaviour cloning loss to reuse samples from past experiences. In contrast to SIL, AWR is a standalone algorithm, and does not need to be combined with an auxiliary RL algorithm. Neumann & Peters (2009) proposed LAWER, a kernel-based fitted Q-iteration algorithm where the Bellman error is weighted by the normalized advantage of each state-action pair. This was then followed by a soft-policy improvement step. Similar to Neumann & Peters (2009), our method also uses exponentiated advantages during policy updates, but their definition of the policy is different from the one in AWR and does not enforce a trust region constraint. Furthermore, AWR does not perform fitted Q-iteration, and instead utilizes off-policy data in a simple constrained policy search procedure. Wang et al. (2018) applied a similar advantage-weighting scheme for imitation learning, but the method was not demonstrated for the RL setting. In this work, we propose several design

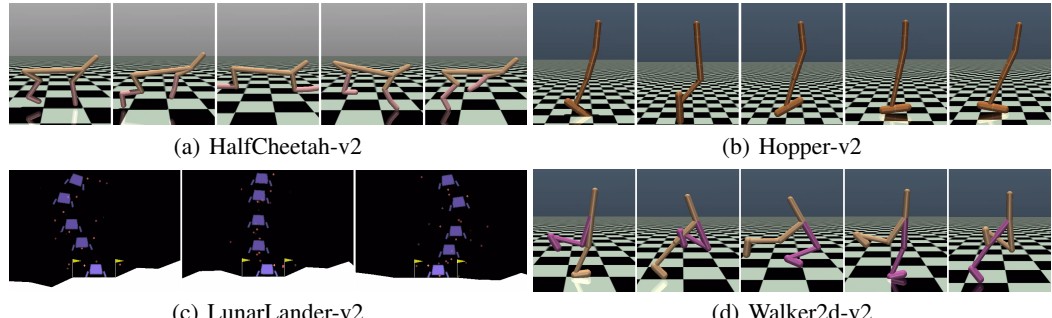

(a) HalfCheetah-v2            (b) Hopper-v2

(c) LunarLander-v2            (d) Walker2d-v2

Figure 2: Snapshots of AWR policies trained on OpenAI Gym tasks. Our simple algorithm learns effective policies for a diverse set of discrete and continuous control tasks.

decisions that are vital for an effective RL algorithm. We also provide a theoretical analysis of AWR when combined with experience replay, and show that the algorithm indeed optimizes the expected improvement with respect to a trajectory-level mixture of past policies modeled by a replay buffer.

## 5 EXPERIMENTS

Our experiments aim to comparatively evaluate the performance of AWR to commonly used on-policy and off-policy deep RL algorithms. We evaluate our method on the OpenAI Gym benchmarks (Brockman et al., 2016), consisting of discrete and continuous control tasks. We also evaluate our method on complex motion imitation tasks with high-dimensional simulated characters, including a 34 DoF humanoid and 64 DoF dog (Peng et al., 2018). We then demonstrate the effectiveness of AWR on fully off-policy learning, by training on static datasets of demonstrations collected from demo policies. Behaviors learned by the policies are best seen in the supplementary video[1]. Code for our implementation of AWR is available at sites.google.com/view/awr-supp/. At each iteration, the agent collects a batch of approximately 2000 samples, which are stored in the replay buffer $\mathcal{D}$ along with samples from previous iterations. The replay buffer stores 50k of the most recent samples. Updates to the value function and policy are performed by uniformly sampling minibatches of 256 samples from $\mathcal{D}$. The value function is updated with 200 gradient steps per iteration, and the policy is updated with 1000 steps. Detailed hyperparameter settings are provided in Appendix C.

### 5.1 BENCHMARKS

We compare AWR to a number of state-of-the-art RL algorithms, including on-policy algorithms, such as TRPO (Schulman et al., 2015) and PPO (Schulman et al., 2017), off-policy algorithms, such as DDPG (Lillicrap et al., 2016), TD3 (Fujimoto et al., 2018), and SAC (Haarnoja et al., 2018a), as well as RWR (Peters & Schaal, 2007), which we include for comparison due to its similarity to AWR.[2] TRPO and PPO use the implementations from OpenAI baselines (Dhariwal et al., 2017). DDPG, TD3, and SAC uses the implementations from RLkit (Pong, 2019). RWR is a custom implementation following the algorithm described by Peters & Schaal (2007).

Snapshots of the AWR policies are shown in Figure 2. Learning curves comparing the different algorithms on the OpenAI Gym benchmarks are shown in Figure 3, and Table 1 summarizes the average returns of the final policies across 5 training runs initialized with different random seeds. Overall, AWR shows competitive performance with the state-of-the-art deep RL algorithms. It significantly outperforms on-policy methods such as PPO and TRPO in both sample efficiency and asymptotic performance. While it is not yet as sample efficient as current state-of-the-art off-policy methods, such SAC and TD3, it is generally able to achieve a similar asymptotic performance, despite using only simple supervised regression for both policy and value function updates. The complex Humanoid-V2 task proved to be the most challenging case for AWR, and its performance still lags well behind SAC. Note that RWR generally does not perform well on any of these tasks. This sug-

---

[2]While we attempted to compare to MPO (Abdolmaleki et al., 2018), we were unable to find source code for an implementation that reproduces results comparable to those reported by Abdolmaleki et al. (2018), and could not implement the algorithm such that it achieves similar performance to those reported by the authors.

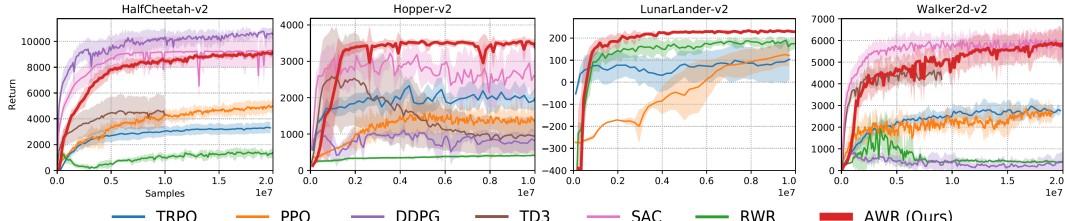

Figure 3: Learning curves of the various algorithms when applied to OpenAI Gym tasks. Results are averaged across 5 random seeds. AWR is generally competitive with the best current methods.

| Task | TRPO | PPO | DDPG | TD3 | SAC | RWR | AWR (Ours) |
|---|---|---|---|---|---|---|---|
| Ant-v2 | $2901 \pm 85$ | $1161 \pm 389$ | $72 \pm 1550$ | $4285 \pm 671$ | $\mathbf{5909 \pm 371}$ | $181 \pm 19$ | $5067 \pm 256$ |
| HalfCheetah-v2 | $3302 \pm 428$ | $4920 \pm 429$ | $\mathbf{10563 \pm 382}$ | $4309 \pm 1238$ | $9297 \pm 1206$ | $1400 \pm 370$ | $9136 \pm 184$ |
| Hopper-v2 | $1880 \pm 337$ | $1391 \pm 304$ | $855 \pm 282$ | $935 \pm 489$ | $2769 \pm 552$ | $605 \pm 114$ | $\mathbf{3405 \pm 121}$ |
| Humanoid-v2 | $552 \pm 9$ | $695 \pm 59$ | $4382 \pm 423$ | $81 \pm 17$ | $\mathbf{8048 \pm 700}$ | $509 \pm 18$ | $4996 \pm 697$ |
| LunarLander-v2 | $104 \pm 94$ | $121 \pm 49$ | $-$ | $-$ | $-$ | $185 \pm 23$ | $\mathbf{229 \pm 2}$ |
| Walker2d-v2 | $2765 \pm 168$ | $2617 \pm 362$ | $401 \pm 470$ | $4212 \pm 427$ | $\mathbf{5805 \pm 587}$ | $406 \pm 64$ | $5813 \pm 483$ |

Table 1: Final returns for different algorithms on the OpenAI Gym tasks, with $\pm$ corresponding to one standard deviation of the average return across 5 random seeds. In terms of final performance, AWR generally performs comparably or better than prior methods.

gests that, although AWR is simple and easy to implement, the particular modifications it makes compared to standard RWR are critical for effective performance. To illustrate AWR's generality on tasks with discrete actions, we compare AWR to TRPO, PPO, and RWR on LunarLander-v2. DDPG, TD3, and SAC are not easily applicable to discrete action spaces due to their need to back-propagate from the Q-function to the policy. On this discrete control task, AWR also shows strong performance compared to the other algorithms.

## 5.2 ABLATION EXPERIMENTS

To determine the effects of various design decisions, we evaluate the performance of AWR when key components of the algorithm have been removed. The experiments include an on-policy version of AWR (On-Policy), where only data collected from the latest policy is used to perform updates. We also compare with a version of AWR without the baseline $V(\mathbf{s})$ (No Baseline), which corresponds to using the standard RWR weights $\omega_{\mathbf{s},\mathbf{a}} = \exp(\frac{1}{\beta}\mathcal{R}_{\mathbf{s},\mathbf{a}})$, and another version that uses Monte Carlo return estimates instead of TD($\lambda$) (No TD($\lambda$)). The effects of these components are illustrated in Figure 4. Overall, these design decisions appear to be vital for an effective algorithm, with the most crucial components being the use of experience replay and a baseline. Updates using only on-policy data can lead to instabilities and result in noticeable degradation in performance, which may be due to overfitting on a smaller dataset. This issue might be mitigated by collecting a larger batch of on-policy data per iteration, but this can also negatively impact sample efficiency. Removing the baseline also noticeably hampers performance. Using simple Monte Carlo return estimates instead of TD($\lambda$) seems to be a viable alternative, and the algorithm still achieves competitive performance on some tasks. When combined, these different components yield substantial performance gains over standard RWR.

To better evaluate the effect of experience replay on AWR, we compare the performance of policies trained with different capacities for the replay buffer. Figure 4 illustrates the learning curves for buffers of size 5k, 20k, 50k, 100k, and 500k, with 50k being the default buffer size in our experiments. The size of the replay buffer appears to have a significant impact on overall performance. Smaller buffer sizes can result in instabilities during training, which again may be an effect of overfitting to a smaller dataset. As the buffer size increases, AWR remains stable even when the dataset is dominated by off-policy data from previous iterations. In fact, performance over the course of training appears more stable with larger replay buffers, but progress can also become slower. Since the sampling policy $\mu(\mathbf{a}|\mathbf{s})$ is modeled by the replay buffer, a larger buffer can limit the rate at which $\mu$ changes by maintaining older data for more iterations. Due to the trust region penalty in Equation 7, a slower changing $\mu$ also prevents the policy $\pi$ from changing quickly. The replay buffer therefore provides a simple mechanism to trade-off between stability and learning speed.

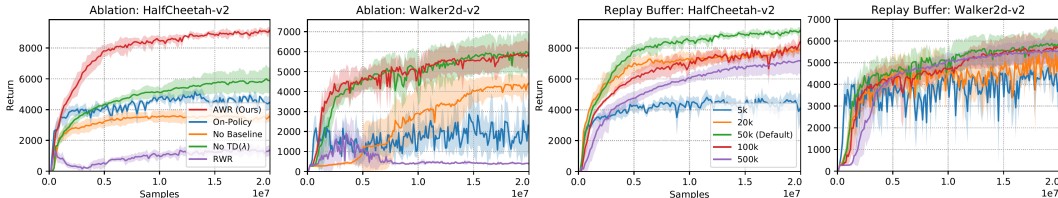

Figure 4: **Left:** Learning curves comparing AWR with various components removed. Each component appears to contribute to improvements in performance, with the best performance achieved when all components are combined. **Right:** Learning curves comparing AWR with different capacity replay buffers. AWR remains stable with large replay buffers containing primarily off-policy data from previous iterations of the algorithm.

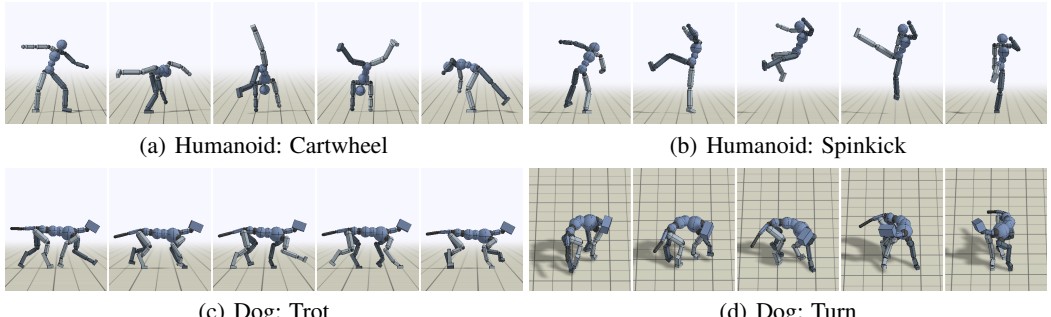

(a) Humanoid: Cartwheel      (b) Humanoid: Spinkick

(c) Dog: Trot      (d) Dog: Turn

Figure 5: Snapshots of 34 DoF humanoid and 64 DoF dog trained with AWR to imitate reference motion recorded from real world subjects. AWR is able to learn sophisticated skills with characters with large numbers of degrees of freedom.

## 5.3 MOTION IMITATION

The Gym benchmarks present relatively low-dimensional tasks. In this section, we study how AWR can solve higher-dimensional tasks with complex simulated characters, including a 34 DoF humanoid and 64 DoF dog. The objective of the tasks is to imitate reference motion clips recorded using motion capture from real world subjects. The experimental setup follows the motion imitation framework proposed by Peng et al. (2018). Motion clips are collected from publicly available datasets (CMU; SFU; Zhang et al., 2018). The skills include highly dynamics motions, such as spinkicks and canters (i.e. running), and motions that requires more coordinated movements of the character's body, such as a cartwheel. Snapshots of the behaviors learned by the AWR policies are available in Figure 5. Table 2 compares the performance of AWR to RWR and the highly-tuned PPO implementation from Peng et al. (2018). Learning curves for the different algorithms are shown in Figure 6. AWR performs well across the set of challenging skills, consistently achieving comparable or better performance than PPO. RWR struggles with controlling the humanoid, but exhibits stronger performance on the dog. This performance difference may be due to the more dynamic and acrobatic skills of the humanoid, compared to the more standard locomotion skills of the dog.

## 5.4 OFF-POLICY LEARNING WITH STATIC DATASETS

Since AWR is an off-policy RL algorithm, it has the advantage of being able to leverage data from other sources. This not only accelerates the learning process on standard tasks, as discussed above, but also allows us to apply AWR in a fully off-policy setting, where the algorithm is provided with a static dataset of transitions, and then tasked with learning the best possible policy. To evaluate our method in this setting, we use the off-policy tasks proposed by Kumar et al. (2019). The objective of these tasks is to learn policies solely from static datasets, without collecting any additional data from the policy that is being trained. The dataset consists of trajectories $\tau = \{(s_0, a_0, r_0), (s_1, a_1, r_1), ...\}$ from rollouts of a demo policy. Unlike standard imitation learning tasks, which only observes the states and actions from the demo policy, the dataset also provides the reward received by the demo policy at each step. The demo policies are trained using SAC on various OpenAI Gym tasks. A dataset of 1 million timesteps is collected for each task.

| Task | PPO | RWR | AWR (Ours) |
|---|---|---|---|
| Humanoid: Cartwheel | $0.76 \pm 0.02$ | $0.03 \pm 0.01$ | $\mathbf{0.78 \pm 0.07}$ |
| Humanoid: Spinkick | $0.70 \pm 0.02$ | $0.05 \pm 0.03$ | $\mathbf{0.77 \pm 0.04}$ |
| Dog: Canter | $0.76 \pm 0.03$ | $0.78 \pm 0.04$ | $\mathbf{0.86 \pm 0.01}$ |
| Dog: Trot | $\mathbf{0.86 \pm 0.01}$ | $\mathbf{0.86 \pm 0.01}$ | $\mathbf{0.86 \pm 0.03}$ |
| Dog: Turn | $0.75 \pm 0.02$ | $0.75 \pm 0.03$ | $\mathbf{0.82 \pm 0.03}$ |

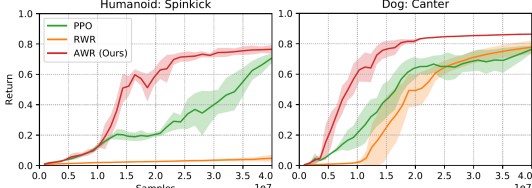

Table 2: Performance statistics of algorithms on the motion imitation tasks. Returns are normalized between the minimum and maximum possible returns per episode.

Figure 6: Learning curves on motion imitation tasks. On these challenging tasks, AWR generally learns faster than PPO and RWR.

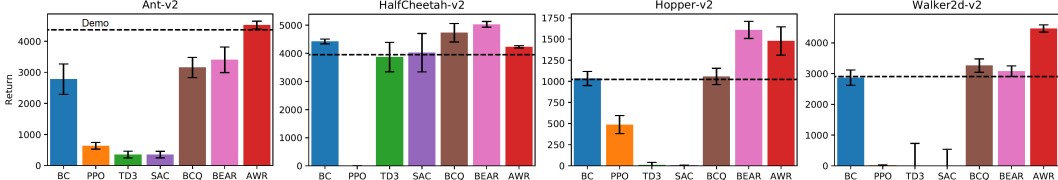

Figure 7: Performance of various algorithms on off-policy learning tasks with static datasets. AWR is able to learn policies that are comparable or better than the original demo policies.

For AWR, we simply treat the dataset as the replay buffer $\mathcal{D}$ and directly apply the algorithm without additional modifications. Figure 7 compares AWR with other algorithms when applied to the datasets. We include comparisons to the performance of the original demo policy used to generate the dataset (Demo) and a behavioral cloning policy (BC). The comparisons also include recent off-policy methods: batch-constrained Q-learning (BCQ) (Fujimoto et al., 2019) and bootstrapping error accumulation reduction (BEAR) (Kumar et al., 2019), which have shown strong performance on off-policy learning with static datasets. Note that both of these prior methods are modifications to existing off-policy RL methods, such as TD3 and SAC, which are already quite complex. In contrast, AWR is simple and requires no modifications for the fully off-policy setting. Despite not collecting any additional data, AWR is able to learn effective policies from these fully off-policy datasets, achieving comparable or better performance than the original demo policies. On-policy methods, such as PPO performs poorly in this off-policy setting. Q-function based methods, such as TD3 and SAC, can in principle handle off-policy data but, as discussed in prior work, tend to struggle in this setting in practice (Fujimoto et al., 2019; Kumar et al., 2019). Indeed, standard behavioral cloning (BC) often outperforms these standard RL methods. In this fully off-policy setting, AWR can be interpreted as an *advantage-weighted* form of behavioral cloning, which assigns higher likelihoods to demonstration actions that receive higher advantages. Unlike Q-function based methods, AWR is less susceptible to issues from out-of-distribution actions as the policy is always trained on observed actions from the behaviour data (Kumar et al., 2019). AWR also shows comparable performance to BEAR and BCQ, which are specifically designed for this off-policy setting and introduce considerable algorithmic overhead.

## 6 DISCUSSION AND FUTURE WORK

We presented advantage-weighted regression, a simple off-policy reinforcement learning algorithm, where policy updates are performed using standard supervised learning methods. Despite its simplicity, our algorithm is able to solve challenging control tasks with complex simulated agents, and achieve competitive performance on standard benchmarks compared to a number of well-established RL algorithms. Our derivation introduces several new design decisions, and our experiments verify the importance of these components. AWR is also able to learn from fully off-policy datasets, demonstrating comparable performance to state-of-the-art off-policy methods. While AWR is effective for a diverse suite of tasks, it is not yet as sample efficient as the most efficient off-policy algorithms. We believe that exploring techniques for improving sample efficiency and performance on fully off-policy learning can open opportunities to deploy these methods in real world domains. We are also interested in exploring applications that are particularly suitable for these regression-based RL algorithms, as compared to other classes of RL techniques. A better theoretical understanding of the convergence properties of these algorithms, especially when combined with experience replay, could also be valuable for the development of future algorithms.

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

## A AWR Derivation

In this section, we derive the AWR algorithm as an approximate optimization of a constrained policy search problem. Our goal is to find a policy that maximize the expected *improvement* $\eta(\pi) = J(\pi) - J(\mu)$ over a sampling policy $\mu(\mathbf{a}|\mathbf{s})$. We start with a lemma from Kakade & Langford (2002), which shows that the expected improvement can be expressed in terms of the advantage $A^\mu(\mathbf{s}, \mathbf{a}) = \mathcal{R}^\mu_{\mathbf{s},\mathbf{a}} - V^\mu(\mathbf{s})$ with respect to the sampling policy $\mu$, where $\mathcal{R}^\mu_{\mathbf{s},\mathbf{a}}$ denotes the return obtained by performing action $\mathbf{a}$ in state $\mathbf{s}$ and following $\mu$ for the following timesteps, and $V^\mu(\mathbf{s}) = \int_{\mathbf{a}} \mu(\mathbf{a}|\mathbf{s})\mathcal{R}^{\mathbf{a}}_{\mathbf{s}}\, d\mathbf{a}$ corresponds to the value function of $\mu$,

$$\mathbb{E}_{\tau \sim p_\pi(\tau)} \left[ \sum_{t=0}^{\infty} \gamma^t A^\mu(\mathbf{s}_t, \mathbf{a}_t) \right] \tag{17}$$

$$= \mathbb{E}_{\tau \sim p_\pi(\tau)} \left[ \sum_{t=0}^{\infty} \gamma^t \left( r(\mathbf{s}_t, \mathbf{a}_t) + \gamma V^\mu(\mathbf{s}_{t+1}) - V^\mu(\mathbf{s}_t) \right) \right] \tag{18}$$

$$= \mathbb{E}_{\tau \sim p_\pi(\tau)} \left[ -V^\mu(\mathbf{s}_0) + \sum_{t=0}^{\infty} \gamma^t r(\mathbf{s}_t, \mathbf{a}_t) \right] \tag{19}$$

$$= -\mathbb{E}_{\mathbf{s}_0 \sim p(\mathbf{s}_0)} \left[ V^\mu(\mathbf{s}_0) \right] + \mathbb{E}_{\tau \sim p_\pi(\tau)} \left[ \sum_{t=0}^{\infty} \gamma^t r(\mathbf{s}_t, \mathbf{a}_t) \right] \tag{20}$$

$$= -J(\mu) + J(\pi) \tag{21}$$

We can rewrite Equation 22 with an expectation over states instead of trajectories:

$$\eta(\pi) = \mathbb{E}_{\tau \sim p_\pi(\tau)} \left[ \sum_{t=0}^{\infty} \gamma^t A^\mu(\mathbf{s}_t, \mathbf{a}_t) \right] \tag{22}$$

$$= \sum_{t=0}^{\infty} \int_{\mathbf{s}} p(\mathbf{s}_t = \mathbf{s}|\pi) \int_{\mathbf{a}} \pi(\mathbf{a}|\mathbf{s}) \gamma^t A^\mu(\mathbf{s}, \mathbf{a})\, d\mathbf{a}\, d\mathbf{s} \tag{23}$$

$$= \int_{\mathbf{s}} \sum_{t=0}^{\infty} \gamma^t p(\mathbf{s}_t = \mathbf{s}|\pi) \int_{\mathbf{a}} \pi(\mathbf{a}|\mathbf{s}) A^\mu(\mathbf{s}, \mathbf{a})\, d\mathbf{a}\, d\mathbf{s} \tag{24}$$

$$= \int_{\mathbf{s}} d_\pi(\mathbf{s}) \int_{\mathbf{a}} \pi(\mathbf{a}|\mathbf{s}) \left[ \mathcal{R}^\mu_{\mathbf{s},\mathbf{a}} - V^\mu(\mathbf{s}) \right]\, d\mathbf{a}\, d\mathbf{s}, \tag{25}$$

where $d_\pi(\mathbf{s}) = \sum_{t=0}^{\infty} \gamma^t p(\mathbf{s}_t = \mathbf{s}|\pi)$ represents the unnormalized discounted state distribution induced by the policy $\pi$ (Sutton & Barto, 1998), and $p(\mathbf{s}_t = \mathbf{s}|\pi)$ is the likelihood of the agent being in state $\mathbf{s}$ after following $\pi$ for $t$ timesteps.

The objective in Equation 25 can be difficult to optimize due to the dependency between $d_\pi(\mathbf{s})$ and $\pi$, as well as the need to collect samples from $\pi$. Following Schulman et al. (2015), we can optimize an approximation $\hat{\eta}(\pi)$ of $\eta(\pi)$ using the state distribution of $\mu$,

$$\hat{\eta}(\pi) = \int_{\mathbf{s}} d_\mu(\mathbf{s}) \int_{\mathbf{a}} \pi(\mathbf{a}|\mathbf{s}) \left[ \mathcal{R}^\mu_{\mathbf{s},\mathbf{a}} - V^\mu(\mathbf{s}) \right]\, d\mathbf{a}\, d\mathbf{s}. \tag{26}$$

$\hat{\eta}(\pi)$ matches $\eta(\pi)$ to first order (Kakade & Langford, 2002), and provides a reasonable estimate of $\eta$ if $\pi$ and $\mu$ are similar. Using this objective, we can formulate the following *constrained* policy search problem:

$$\arg\max_{\pi} \quad \int_{\mathbf{s}} d_\mu(\mathbf{s}) \int_{\mathbf{a}} \pi(\mathbf{a}|\mathbf{s}) \left[ \mathcal{R}^\mu_{\mathbf{s},\mathbf{a}} - V^\mu(\mathbf{s}) \right]\, d\mathbf{a}\, d\mathbf{s} \tag{27}$$

$$\text{s.t.} \quad \mathrm{D}_{\mathrm{KL}} \left( \pi(\cdot|\mathbf{s}) || \mu(\cdot|\mathbf{s}) \right) \leq \epsilon, \quad \forall\, \mathbf{s} \tag{28}$$

$$\int_{\mathbf{a}} \pi(\mathbf{a}|\mathbf{s})\, d\mathbf{a} = 1, \quad \forall\, \mathbf{s}. \tag{29}$$

Since enforcing the pointwise KL constraint in Equation 28 at all states is intractable, we relax the constraint by enforcing it only in expectation $\int_{\mathbf{s}} d_\mu(\mathbf{s}) \mathrm{D}_{\mathrm{KL}} \left( \pi(\cdot|\mathbf{s}) || \mu(\cdot|\mathbf{s}) \right) d\mathbf{s} \leq \epsilon$. To further

simplify the optimization problem, we relax the hard KL constraint by converting it into a soft constraint with coefficient $\beta$,

$$\arg\max_{\pi} \quad \left( \int_{\mathbf{s}} d_{\mu}(\mathbf{s}) \int_{\mathbf{a}} \pi(\mathbf{a}|\mathbf{s}) \left[ \mathcal{R}_{\mathbf{s},\mathbf{a}}^{\mu} - V^{\mu}(\mathbf{s}) \right] \, d\mathbf{a} \, d\mathbf{s} \right) + \beta \left( \epsilon - \int_{\mathbf{s}} d_{\mu}(\mathbf{s}) \mathrm{D}_{\mathrm{KL}} \left( \pi(\cdot|\mathbf{s}) || \mu(\cdot|\mathbf{s}) \right) d\mathbf{s} \right)$$

$$\text{s.t.} \quad \int_{\mathbf{a}} \pi(\mathbf{a}|\mathbf{s}) \, d\mathbf{a} = 1, \quad \forall \, \mathbf{s}.$$

$$(30)$$

Next we form the Lagrangian,

$$\mathcal{L}(\pi, \beta, \alpha) = \left( \int_{\mathbf{s}} d_{\mu}(\mathbf{s}) \int_{\mathbf{a}} \pi(\mathbf{a}|\mathbf{s}) \left[ \mathcal{R}_{\mathbf{s},\mathbf{a}}^{\mu} - V^{\mu}(\mathbf{s}) \right] \, d\mathbf{a} \, d\mathbf{s} \right) + \beta \left( \epsilon - \int_{\mathbf{s}} d_{\mu}(\mathbf{s}) \mathrm{D}_{\mathrm{KL}} \left( \pi(\cdot|\mathbf{s}) || \mu(\cdot|\mathbf{s}) \right) d\mathbf{s} \right)$$

$$+ \int_{\mathbf{s}} \alpha_{\mathbf{s}} \left( 1 - \int_{\mathbf{a}} \pi(\mathbf{a}|\mathbf{s}) d\mathbf{a} \right) d\mathbf{s},$$

$$(31)$$

with $\beta$ and $\alpha = \{\alpha_{\mathbf{s}} \,|\, \forall \mathbf{s} \in \mathcal{S}\}$ corresponding to the Lagrange multipliers. Differentiating $\mathcal{L}(\pi, \beta, \alpha)$ with respect to $\pi(\mathbf{a}|\mathbf{s})$ results in

$$\frac{\partial \mathcal{L}}{\partial \pi(\mathbf{a}|\mathbf{s})} = d_{\mu}(\mathbf{s}) \left( \mathcal{R}_{\mathbf{s},\mathbf{a}}^{\mu} - V^{\mu}(\mathbf{s}) \right) - \beta \, d_{\mu}(\mathbf{s}) \log \pi(\mathbf{a}|\mathbf{s}) + \beta d_{\mu}(\mathbf{s}) \log \mu(\mathbf{a}|\mathbf{s}) - \beta d_{\mu}(\mathbf{s}) - \alpha_{\mathbf{s}}.$$

$$(32)$$

Setting to zero and solving for $\pi(\mathbf{a}|\mathbf{s})$ gives

$$\log \pi(\mathbf{a}|\mathbf{s}) = \frac{1}{\beta} \left( \mathcal{R}_{\mathbf{s},\mathbf{a}}^{\mu} - V^{\mu}(\mathbf{s}) \right) + \log \mu(\mathbf{a}|\mathbf{s}) - 1 - \frac{1}{d_{\mu}(\mathbf{s})} \frac{\alpha_{\mathbf{s}}}{\beta} \tag{33}$$

$$\pi(\mathbf{a}|\mathbf{s}) = \mu(\mathbf{a}|\mathbf{s}) \exp \left( \frac{1}{\beta} \left( \mathcal{R}_{\mathbf{s},\mathbf{a}}^{\mu} - V^{\mu}(\mathbf{s}) \right) \right) \exp \left( -\frac{1}{d_{\mu}(\mathbf{s})} \frac{\alpha_{\mathbf{s}}}{\beta} - 1 \right) \tag{34}$$

Since $\int_{\mathbf{a}} \pi(\mathbf{a}|\mathbf{s}) \, d\mathbf{a} = 1$, the second exponential term is the partition function $Z(\mathbf{s})$ that normalizes the conditional action distribution,

$$Z(\mathbf{s}) = \exp \left( \frac{1}{d_{\mu}(\mathbf{s})} \frac{\alpha_{\mathbf{s}}}{\beta} + 1 \right) = \int_{\mathbf{a}'} \mu(\mathbf{a}'|\mathbf{s}) \exp \left( \frac{1}{\beta} \left( \mathcal{R}_{\mathbf{s},\mathbf{a}'}^{\mu} - V^{\mu}(\mathbf{s}) \right) \right) d\mathbf{a}'. \tag{35}$$

The optimal policy is therefore given by,

$$\pi^*(\mathbf{a}|\mathbf{s}) = \frac{1}{Z(\mathbf{s})} \mu(\mathbf{a}|\mathbf{s}) \exp \left( \frac{1}{\beta} \left( \mathcal{R}_{\mathbf{s},\mathbf{a}}^{\mu} - V^{\mu}(\mathbf{s}) \right) \right) \tag{36}$$

If $\pi$ is represented by a function approximator, the optimal policy $\pi^*$ can be projected onto the manifold of parameterized policies by solving the following supervised regression problem

$$\arg\min_{\pi} \quad \mathbb{E}_{\mathbf{s} \sim d_{\mu}(\mathbf{s})} \left[ \mathrm{D}_{\mathrm{KL}} \left( \pi^*(\cdot|\mathbf{s}) || \pi(\cdot|\mathbf{s}) \right) \right] \tag{37}$$

$$= \arg\min_{\pi} \quad \mathbb{E}_{\mathbf{s} \sim d_{\mu}(\mathbf{s})} \left[ \mathrm{D}_{\mathrm{KL}} \left( \frac{1}{Z(\mathbf{s})} \mu(\mathbf{a}|\mathbf{s}) \exp \left( \frac{1}{\beta} \left( \mathcal{R}_{\mathbf{s},\mathbf{a}}^{\mu} - V^{\mu}(\mathbf{s}) \right) \right) \middle|\middle| \pi(\cdot|\mathbf{s}) \right) \right] \tag{38}$$

$$= \arg\max_{\pi} \quad \mathbb{E}_{\mathbf{s} \sim d_{\mu}(\mathbf{s})} \mathbb{E}_{\mathbf{a} \sim \mu(\mathbf{a}|\mathbf{s})} \left[ \log \pi(\mathbf{a}|\mathbf{s}) \exp \left( \frac{1}{\beta} \left( \mathcal{R}_{\mathbf{s},\mathbf{a}}^{\mu} - V^{\mu}(\mathbf{s}) \right) \right) \right], \tag{39}$$

## B  AWR DERIVATION WITH EXPERIENCE REPLAY

In this section, we extend the derivation presented in Appendix A to incorporate experience replay using a replay buffer containing data from previous policies. To recap, the sampling distribution is a mixture of $k$ past policies $\{\pi_1, \cdots, \pi_k\}$, where the mixture is performed at the trajectory level. First, we define the trajectory distribution $\mu(\tau)$, marginal state-action distribution $\mu(\mathbf{s}, \mathbf{a})$, and marginal state distribution $d_{\mu}(\mathbf{s})$ of the replay buffer according to:

$$\mu(\tau) = \sum_{i=1}^{k} w_i d_{\pi_i}(\tau), \qquad \mu(\mathbf{s}, \mathbf{a}) = \sum_{i=1}^{k} w_i d_{\pi_i}(\mathbf{s}) \pi_i(\mathbf{a}|\mathbf{s}), \quad d_{\mu}(\mathbf{s}) = \sum_{i=1}^{k} w_i d_{\pi_i}(\mathbf{s}) \tag{40}$$

where the weights $\sum_i w_i = 1$ specify the probabilities of selecting each policy $\pi_i$. The conditional action distribution $\mu(\mathbf{a}|\mathbf{s})$ induced by the replay buffer is given by:

$$\mu(\mathbf{a}|\mathbf{s}) = \frac{\mu(\mathbf{s}, \mathbf{a})}{d_\mu(\mathbf{s})} = \frac{\sum_{i=1}^{k} w_i d_{\pi_i}(\mathbf{s}) \pi_i(\mathbf{a}|\mathbf{s})}{\sum_{j=1}^{k} w_j d_{\pi_j}(\mathbf{s})}. \tag{41}$$

Next, using Lemma 6.1 from Kakade & Langford (2002) (also derived in Appendix A), the expected improvement of $\pi$ over each policy $\pi_i$ satisfies

$$J(\pi) = J(\pi_i) + \mathbb{E}_{\mathbf{s} \sim d_\pi(\mathbf{s}), a \sim \pi(\mathbf{a}|\mathbf{s})} \left[ A^{\pi_i}(\mathbf{s}, \mathbf{a}) \right] \tag{42}$$

The expected improvement over the mixture can then be expressed with respect to the individual policies,

$$\eta(\pi) = J(\pi) - J(\mu) \tag{43}$$

$$= J(\pi) - \sum_{i=1}^{k} w_i J(\pi_i) \tag{44}$$

$$= \sum_{i=1}^{k} w_i \left( J(\pi) - J(\pi_i) \right) \tag{45}$$

$$= \sum_{i=1}^{k} w_i \left( \mathbb{E}_{\mathbf{s} \sim d_\pi(\mathbf{s}), \mathbf{a} \sim \pi(\mathbf{a}|\mathbf{s})} \left[ A^{\pi_i}(\mathbf{s}, \mathbf{a}) \right] \right) \tag{46}$$

In order to ensure that the policy $\pi$ is similar to the past policies, we constrain $\pi$ against the conditional action distributions of the replay buffer,

$$\mathbb{E}_{\mathbf{s} \sim \mu(\mathbf{s})} \left[ D_{KL} \left( \pi(\mathbf{a}|\mathbf{s}) \big|\big| \mu(\mathbf{a}|\mathbf{s}) \right) \right] \leq \varepsilon. \tag{47}$$

Note that constraining $\pi$ against $\mu(\mathbf{a}|\mathbf{s})$ has a number of desirable properties. First, the constraint prevents the policy $\pi$ from choosing actions that are vastly different from **all** of the policies $\{\pi_1, \cdots, \pi_k\}$. Second, the mixture weight assigned to each $\pi_i$ in the definition of $\mu$ depends on the marginal state density $d_{\pi_i}(\mathbf{s})$ for the particular policy. This property is desirable as the policy $\pi$ is now constrained to be similar to $\pi_i$ only at states that are likely to be visited by $\pi_i$. This then yields the following constrained objective:

$$\arg\max_{\pi} \quad \sum_{i=1}^{k} w_i \, \mathbb{E}_{\mathbf{s} \sim d_{\pi_i}(\mathbf{s})} \mathbb{E}_{\mathbf{a} \sim \pi(\mathbf{a}|\mathbf{s})} \left[ \mathcal{R}_{\mathbf{s},\mathbf{a}}^{\pi_i} - V^{\pi_i}(\mathbf{s}) \right] \tag{48}$$

$$\text{s.t.} \quad \mathbb{E}_{\mathbf{s} \sim d_\mu(\mathbf{s})} \left[ D_{KL} \left( \pi(\cdot|\mathbf{s}) || \mu(\cdot|\mathbf{s}) \right) \right] \leq \epsilon, \tag{49}$$

$$\int_{\mathbf{a}} \pi(\mathbf{a}|\mathbf{s}) \, d\mathbf{a} = 1, \quad \forall \, \mathbf{s}. \tag{50}$$

The Lagrangian of the above objective is given by:

$$\mathcal{L}(\pi, \beta, \alpha) = \left( \sum_i w_i \mathbb{E}_{\mathbf{s} \sim d_{\pi_i}(\mathbf{s})} \mathbb{E}_{\mathbf{a} \sim \pi(\mathbf{a}|\mathbf{s})} \left[ \mathcal{R}_{\mathbf{s},\mathbf{a}}^{\pi_i} - V^{\pi_i}(\mathbf{s}) \right] \right)$$

$$+ \beta \left( \epsilon - \mathbb{E}_{\mathbf{s} \sim d_\mu(\mathbf{s})} D_{KL} \left( \pi(\cdot|\mathbf{s}) \Bigg|\Bigg| \frac{\sum_{i=1}^{k} w_i d_{\pi_i}(\mathbf{s}) \pi_i(\cdot|\mathbf{s})}{\sum_{j=1}^{k} w_j d_{\pi_j}(\mathbf{s})} \right) \right) \tag{51}$$

$$+ \int_{\mathbf{s}} \alpha_{\mathbf{s}} \left( 1 - \int_{\mathbf{a}} \pi(\mathbf{a}|\mathbf{s}) d\mathbf{a} \right) d\mathbf{s},$$

Solving the Lagrangian following the same procedure as Appendix A leads to an optimal policy of the following form:

$$\pi^*(\mathbf{a}|\mathbf{s}) = \frac{1}{Z(\mathbf{s})} \mu(\mathbf{a}|\mathbf{s}) \exp \left( \frac{1}{\beta} \frac{\sum_i w_i d_{\pi_i}(\mathbf{s}) \left( \mathcal{R}_{\mathbf{s},\mathbf{a}}^{\pi_i} - V^{\pi_i}(\mathbf{s}) \right)}{\sum_j w_j d_{\pi_j}(\mathbf{s})} \right) \tag{52}$$

Finally, if $\pi$ is represented by a function approximator, the optimal policy $\pi^*$ can be projected onto the manifold of parameterized policies by solving the following supervised regression problem

$$\arg\min_{\pi} \quad \mathbb{E}_{\mathbf{s},\sim d_\mu(\mathbf{s})} \left[ D_{\mathrm{KL}} \left( \pi^*(\cdot|\mathbf{s}) || \pi(\cdot|\mathbf{s}) \right) \right] \tag{53}$$

$$= \arg\min_{\pi} \quad \mathbb{E}_{\mathbf{s}\sim d_\mu(\mathbf{s})} \left[ D_{\mathrm{KL}} \left( \frac{1}{Z(\mathbf{s})} \mu(\mathbf{a}|\mathbf{s}) \exp\left( \frac{1}{\beta} \frac{\sum_i w_i d_{\pi_i}(\mathbf{s}) \left( \mathcal{R}^{\pi_i}_{\mathbf{s},\mathbf{a}} - V^{\pi_i}(\mathbf{s}) \right)}{\sum_j w_j d_{\pi_j}(\mathbf{s})} \right) \middle\| \pi(\cdot|\mathbf{s}) \right) \right] \tag{54}$$

One of the challenges of optimizing the objective in Equation 54 is that computing the expected return in the exponent requires rolling out multiple policies starting from the same state, which would require the environment to be resettable to any given state. Therefore, to obtain a more practical objective, we approximate the expected return across policies using a single rollout from the replay buffer,

$$\frac{\sum_i w_i d_{\pi_i}(\mathbf{s}) \mathcal{R}^{\pi_i}_{\mathbf{s},\mathbf{a}}}{\sum_j w_j d_{\pi_j}(\mathbf{s})} \approx \mathcal{R}^{\mathcal{D}}_{\mathbf{s},\mathbf{a}} \text{ such that } (\mathbf{s},\mathbf{a}) \in \mathcal{D} \tag{55}$$

This single-sample estimator results in a biased estimate of the exponentiated advantage, because the expectation with respect to the mixture weights appears in the exponent. But in practice, we find this biased estimator to be effective for our experiments. Therefore, the objective used in practice is given by:

$$\arg\max_{\pi} \quad \sum_{i=1}^{k} w_i \, \mathbb{E}_{\mathbf{s}\sim d_{\pi_i}(\mathbf{s})} \mathbb{E}_{\mathbf{a}\sim\pi_i(\mathbf{a}|\mathbf{s})} \left[ \log \pi(\mathbf{a}|\mathbf{s}) \exp\left( \frac{1}{\beta} \left( \mathcal{R}^{\pi_i}_{\mathbf{s},\mathbf{a}} - \frac{\sum_j w_j d_{\pi_j}(\mathbf{s}) V^{\pi_j}(\mathbf{s})}{\sum_j w_j d_{\pi_j}(\mathbf{s})} \right) \right) \right], \tag{56}$$

where the expectations can be approximated by simply sampling from $\mathcal{D}$ following Line 6 of Algorithm 1. Note, the baseline in the exponent now consists of an average of the value functions of the different policies. One approach for estimating this quantity would be to fit separate value functions $V^{\pi_i}$ for each policy. However, if only a small amount of data is available from each policy, then $V^{\pi_i}$ could be highly inaccurate. Therefore, instead of learning separate value functions, we fit a single *mean* value function $\bar{V}(\mathbf{s})$ that directly estimates the weighted average of $V^{\pi_i}$'s,

$$\bar{V} = \arg\min_{V} \sum_i w_i \, \mathbb{E}_{\mathbf{s},\sim d_{\pi_i}(\mathbf{s})} \mathbb{E}_{\mathbf{a}\sim\pi_i(\mathbf{a}|\mathbf{s})} \left[ ||\mathcal{R}^{\pi_i}_{\mathbf{s},\mathbf{a}} - V(\mathbf{s})||^2 \right] \tag{57}$$

This loss can also be approximated by simply sampling from the replay buffer following Line 5 of Algorithm 1. The optimal solution $\bar{V}(\mathbf{s}) = \frac{\sum_i w_i d_{\pi_i}(\mathbf{s}) V^{\pi_i}(\mathbf{s})}{\sum_j w_j d_{\pi_j}(\mathbf{s})}$ is exactly the baseline in Equation 56.

## C  EXPERIMENTAL SETUP

In our experiments, the policy is represented by a fully-connected network with 2 hidden layers consisting of 128 and 64 ReLU units respectively (Nair & Hinton, 2010), followed by a linear output layer. The value function is modeled by a separate network with a similar architecture, but consists of a single linear output unit for the value. Stochastic gradient descent with momentum is used to update both the policy and value function. The stepsize of the policy and value function are $5\times10^{-5}$ and $1\times10^{-4}$ respectively, and a momentum of $0.9$ is used for both. The temperature is set to $\beta = 0.05$ for all experiments, and $\lambda = 0.95$ is used for TD($\lambda$). The weight clipping threshold $\omega_{\max}$ is set to 20. At each iteration, the agent collects a batch of approximately 2000 samples, which are stored in the replay buffer $\mathcal{D}$ along with samples from previous iterations. The replay buffer stores 50k of the most recent samples. Updates to the value function and policy are performed by uniformly sampling minibatches of 256 samples from $\mathcal{D}$. The value function is updated with 200 gradient steps per iteration, and the policy is updated with 1000 gradient steps.

# D    SIMILARITIES TO POLICY GRADIENTS

On the surface, the AWR policy update bears striking similarities to a conventional policy gradient (PG) update (Sutton et al., 2000):

$$\mathbb{E}_{\mathbf{s}\sim d_\pi(\mathbf{s})}\mathbb{E}_{\mathbf{a}\sim\pi(\mathbf{a}|\mathbf{s})}\left[\nabla_\pi \log \pi(\mathbf{a}|\mathbf{s})\ \left(\mathcal{R}^\pi_{\mathbf{s},\mathbf{a}} - V^\pi(\mathbf{s})\right)\right] \qquad \text{(Policy Gradient)}$$

$$\mathbb{E}_{\mathbf{s}\sim d_\mu(\mathbf{s})}\mathbb{E}_{\mathbf{a}\sim\mu(\mathbf{a}|\mathbf{s})}\left[\nabla_\pi \log \pi(\mathbf{a}|\mathbf{s}) \exp\left(\frac{1}{\beta}\left(\mathcal{R}^\mu_{\mathbf{s},\mathbf{a}} - V^\mu(\mathbf{s})\right)\right)\right]. \qquad \text{(AWR)}$$

However, there are a number of subtle but important differences between the two. First, basic policy gradient algorithms are on-policy methods, which requires the data to be sampled from the same policy $\pi$ that is being optimized $\mathbf{s}\sim d_\pi(\mathbf{s})$ and $\mathbf{a}\sim\pi(\mathbf{a}|\mathbf{s})$, whereas AWR can in principle learn using data from any sampling distribution $\mu$. This requirement for policy gradient methods is because PG directly differentiates through the sampling distribution to compute the gradient of the expected return with respect to the policy parameters. But with AWR and other EM algorithms, they first construct an estimate of the optimal action distribution at each state, and then projects that action distribution onto the space of parameterized policies. Therefore AWR does not need to differentiate through the sampling distribution, which is a critical feature for settings such as batch RL, where the sampling distribution (e.g. demo policy) may not be available to the agent. In AWR, the log probability of an action $\log\pi(\mathbf{a}|\mathbf{s})$ is weighted by the exponentiated advantage $\exp\left(\frac{1}{\beta}\left(\mathcal{R}_{\mathbf{s},\mathbf{a}} - V(\mathbf{s})\right)\right)$, while in PG the log probability is weighted just by the advantage $\left(\mathcal{R}_{\mathbf{s},\mathbf{a}} - V(\mathbf{s})\right)$ without the exponential. Since the exponentiated advantage is non-negative, the objective used in the AWR update is a maximum likelihood objective that tries to maximize the likelihood of all actions, but to varying amounts depending on the exponentiated advantage. In the case of PG, the advantage can be both positive and negative, therefore PG updates decrease the likelihood of actions with negative advantages, and thus it is not a conventional maximum likelihood objective. In practice, negative TD updates are often a source of instability when applying PG to off-policy data.

# E ADDITIONAL EXPERIMENTS

A comprehensive comparison of AWR with prior methods on all of the tasks considered are available in Figure 8 and 9.

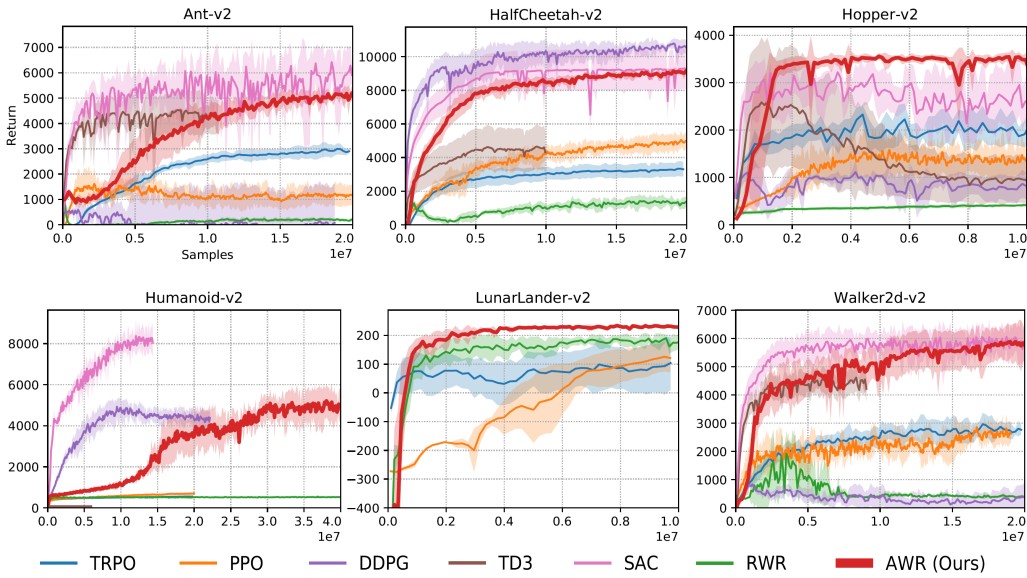

Figure 8: Learning curves of the various algorithms when applied to OpenAI Gym tasks. Results are averaged over 5 random seeds. AWR is generally competitive with the best current methods.

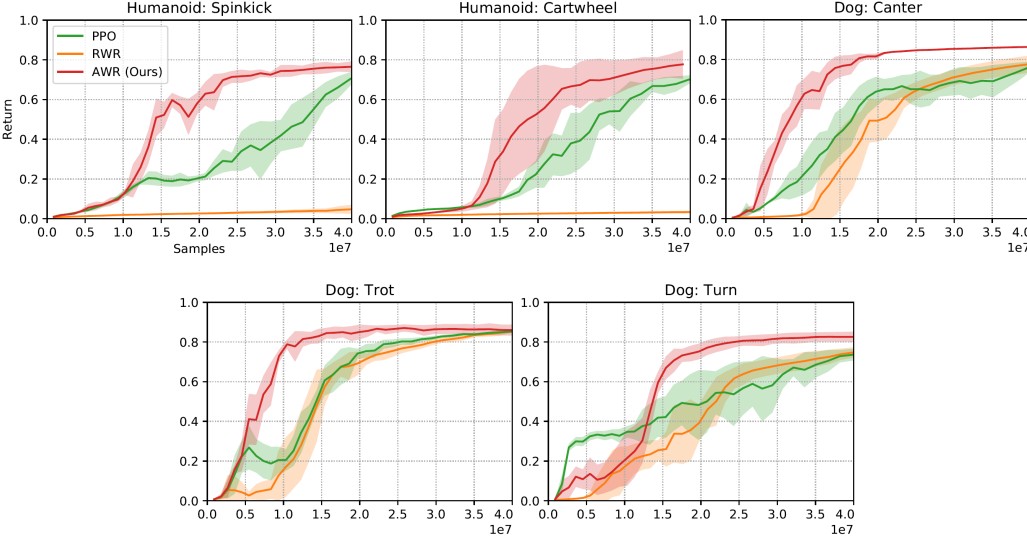

Figure 9: Learning curves on motion imitation tasks. On these challenging tasks, AWR generally learns faster than PPO and RWR.

## E.1 WEIGHT CLIPPING

To analyze the effects of weight clipping on the stability of AWR, we compare learning curves of policies trained with weight clipping using a threshold of $\omega_{\max} = 20$, and policies trained without

weight clipping. Figure 10 compares the learning curves with and without clipping. 5 separate AWR runs with different random seeds are visualized separately. With weight clipping, performance remains stable throughout training. Policies trained without weight clipping are substantially more unstable, exhibiting drastic fluctuations in performance as a result of exploding gradients from excessively large weights. Some training runs without clipping are terminated early due to exploding gradients causing the networks to output NaNs. These experiments suggest that weight clipping is vital for ensuring stable training with AWR.

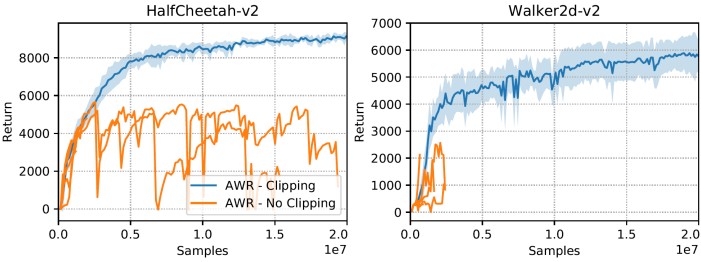

Figure 10: Learning curves comparing AWR policies trained with and without weight clipping. Weight clipping is vital for ensuring stable training with AWR. Policies trained without weight clipping are susceptible to exploding gradients due to excessively large weights.

