# OpenReview forum: "Advantage Weighted Regression: Simple and Scalable Off-Policy Reinforcement Learning"
_ICLR.cc/2020/Conference — Reject_

### Official Review · AnonReviewer3 · 2019-10-07
**Official Blind Review #3**

**Rating:** 6

**Review:**

[Note: I wrote this review after John Schulman's first comment, before any reply, and before Gehrard Neumann's comment]

The authors propose an actor-critic algorithm based mostly on regression. Being off-policy, the algorithm can learn from multiple policies. It can also be applied to continuous as well as to discrete actions, and it can be trained in a batch RL setting. They compare it to a set of state-of-the-art algorithms in standard openAI gym continuous action benchmarks and show competitive performance despite a much simpler implementation of the algorithm.

I basically subscribe to John Schulman's comment below, both about empirical results and about citing Self-Imitation Learning, but I have a stronger point against the paper, which is insufficient positionning with respect to the relevant literature.

The paper does not cite or discuss the one below, though it looks VERY close:

@inproceedings{neumann2009fitted,
  title={Fitted Q-iteration by advantage weighted regression},
  author={Neumann, Gerhard and Peters, Jan R},
  booktitle={Advances in neural information processing systems},
  pages={1177--1184},
  year={2009}
}

This paper also starts from RWR and performs weighted regression based on the advantage rather than on the return. So to me it is exactly the same idea, and it is mandatory that the authors clearly establish what is the novelty of their work with respect to this previous paper.

Less importantly, the authors may also want to have a look at :

@article{zimmer2019exploiting,
  title={Exploiting the sign of the advantage function to learn deterministic policies in continuous domains},
  author={Zimmer, Matthieu and Weng, Paul},
  journal={arXiv preprint arXiv:1906.04556},
  year={2019}
}

which also uses ideas along the same line.

To me, a good way to improve the novelty of this work would be to perform a detailed empirical study of the inner mechanisms of the algorithm on very simple benchmarks where the value function and policy could be visualized. In particular, how stable is the estimation of the value function? This is known to be an issue, as most algorithms avoid approximating it and prefer estimating the Q-function.


**Experience Assessment:**

I have published one or two papers in this area.

**Review Assessment: Checking Correctness Of Derivations And Theory:**

I did not assess the derivations or theory.

**Review Assessment: Checking Correctness Of Experiments:**

I did not assess the experiments.

**Review Assessment: Thoroughness In Paper Reading:**

I read the paper at least twice and used my best judgement in assessing the paper.

---

> ### Author Response · Authors · 2019-11-06
> **clarification**
>
> Thank you for the feedback, we will aim to conduct additional experiments to address your questions. But first, could you clarify what you mean by the stability of the value function? Value functions are very commonly used for actor-critic algorithms, such as policy gradient methods, while Q-functions are more commonly used for off-policy methods. To the best of our knowledge, neither one is necessarily easier to learn than the other.

---

> > ### Comment · AnonReviewer3 · 2019-11-06
> > **Value function and off-policyness**
> >
> > You are right. I have a strong bias towards off-policy methods, which more often use a Q-function, thus my intuition that estimating a value function could be harder than a Q function might be wrong. One recent clue about this intuition was the fact that the value function estimator was removed from the latest version of SAC, another one is that the V function implictly contains a maximization (over Q values) and the presence of this implicit maximization may complicate training (despite a lower dimensionality of V wrt to Q), but I must admit that I have not seen a paper comparing the stability of learning V versus learning Q. Such a paper would be welcome.
> >
> > Anyways, your AWR algorithm pretends to be off-policy, and instability of the critic is a major issue in off-policy algorithms using function approximators, so I'm wondering how your algorithm is behaving in this respect compared to those which use a Q-function.

---

> > > ### Author Response · Authors · 2019-11-07
> > > **value function**
> > >
> > > Thank you for the clarification. Yes, off-policy methods that use only a Q-function to update the policy (e.g. DDPG and SAC) can be more susceptible to model bias in the Q-function, which can lead to instability (https://papers.nips.cc/paper/3964-double-q-learning, https://arxiv.org/abs/1802.09477). For AWR, the policy is  updated using a combination of monte-carlo returns and a value function, where the value function is used primarily as a baseline. This enables AWR to be less susceptible to inaccurate value estimates, compared to methods that use only a value/Q function for policy updates. This can be seen in the experiments from Figure 4. The "No Baseline" policies can be interpreted as using a highly inaccurate value function that just returns zeros for all states. This does lead to some deterioration in performance, but AWR is still able to learn reasonably effective policies. If such an inaccurate value function is used for methods such as DDPG or SAC, the policy will fail to learn anything. Therefore, these results do seem to suggest that AWR can be less susceptible to inaccuracies in the value function.

---

> > > > ### Comment · AnonReviewer3 · 2019-11-08
> > > > **Difference to REINFORCE-like approaches + weight clipping**
> > > >
> > > > Thank you, the reason why AWR is less sensitive to critic approximation errors is now clear.
> > > > By the way, the above simplified description of AWR as a Monte-Carlo-with-baseline method makes it look very close to a REINFORCE-like approach based on the Policy Gradient Theorem (PGT). This led me to the following questions.
> > > >
> > > > REINFORCE-like Monte Carlo approaches using the PG are more stable than bootstrap methods, but they are generally on-policy. My main question is: what is the fundamental point in a weighted regression approach that makes it off-policy? To be even more specific: given that regression is implemented with gradient descent on the residuals and that the loss looks the same as given by the PGT, what is the key difference between the AWR approach and REINFORCE that makes it off-policy? Please forgive me if I'm missing something obvious, I can be quite slow on these matters.
> > > >
> > > > Given that the main weakness of your paper will be lack of novelty (with respect to MARWIL and FQI by AWR), to me the best way to increase the value of the Methods part is to give a much more didactic presentation of the approach than the previous papers.
> > > >
> > > > Another unrelated point: I would be glad to see the impact of weight clipping. An ablation about this implementation detail is missing. Does it drive the algorithm far away from what the derivation suggests?
> > > >
> > > > Note: I wrote this comment before reading your responses to all reviewers.

---

> > > > ### Author Response · Authors · 2019-11-08
> > > > **Response to R3**
> > > >
> > > > Re: Prior work
> > > > We appreciate the pointers to MARWIL and FQI by AWR, and we have included a discussion of this work in the latest revision of the paper (Section 4). We will also perform additional experiments that compare AWR directly with FQI by AWR. We would like to emphasize that FQI by AWR as a method uses a kernel-based Q-function estimator to perform fitted Q-iteration. MARWIL is arguably more related to our method in terms of the form of the update, but addresses a different problem setting -- imitation learning rather than reinforcement learning from scratch.
> > > >
> > > > Re: Improving exposition
> > > > Thank you for the suggestions, we would be happy to revise the presentation to be more didactic. For clarification, could you elaborate on what precisely you have in mind for improving the presentation of the method? We believe that our exposition is already more formal and rigorous than prior work in terms of justifying why we should expect off-policy AWR to perform well. We describe how the method implements a type of trust region (Section 3.1) and how the baseline that give rise to advantages emerges as a natural consequence of the conservative policy improvement objective in Equation 8. We believe that this derivation, which is not presented in Neumann & Peters, contributes substantially in terms of elucidating the reason such a method should work well as an RL algorithm.
> > > >
> > > > Re: difference to PG
> > > > While the AWR update appears similar to a PG update, there are subtle but important differences between the two. In terms of derivation, AWR is not a policy gradient algorithm, it is an EM algorithm (http://is.tuebingen.mpg.de/fileadmin/user_upload/files/publications/ICML2007-Peters_4493[0].pdf). PG methods update a policy by directly estimating the gradient of the expected return with respect to the policy parameters. But with EM algorithms, they first construct an estimate of the optimal action distribution at each state, and then project that action distribution onto the space of parameterized policies. Algorithmically, the two vital differences are:
> > > > 1) in AWR the log probability of an action is weighted by “exp(adv)“, while in PG the log probability is weighted by just “adv” without the exponential. Since exp(adv) is non-negative, the objective used in the AWR update is a maximum likelihood objective that tries to maximize the likelihood of all actions, but to varying amounts depending on exp(adv). In the case of PG, the advantage can be both positive and negative, therefore PG updates will try to decrease the likelihood of actions with a negative advantage and thus it is not a conventional maximum likelihood objective. In practice, negative TD updates are often the source of instability when applying PG to off-policy data.
> > > > 2) policy gradient methods differentiate through the sampling distribution, while AWR can in principle use any off-policy data without the need to differentiate through the sampling distribution (as shown in equation 10).
> > > >
> > > > re: weight clipping
> > > > Yes, we will perform additional ablation experiments to show the impact of weight clipping. We will update you with the results once those are ready.

---

> > > > > ### Comment · AnonReviewer3 · 2019-11-09
> > > > > **Improving exposition**
> > > > >
> > > > > Thanks again for the explanation of the differences to PG, it provides very useful clarifications (that would deserve to be in the paper somehow, together with the explanation of better stability than actor-critic approaches).
> > > > >
> > > > > By the way, I were you, here is how I would improve exposition (I'm not asking, I'm suggesting).
> > > > >
> > > > > First, I would add a "Background" section where I would explain everything that is common to RWR, FQI by AWR (LAWER), MARWIL: the reward weighted approach, the EM approach, and the features of this approach (see just above).
> > > > >
> > > > > Then in the main section I would expose AWR by explaining how it is different from these related algorithms and what additional features it gains from these differences.
> > > > > For instance, the formulas of https://imgur.com/a/0DuRH7o should be given explicitly together with what you said in the related work section about the differences between LAWER and AWR. As is, the non expert reader has to do a lot of work to understand what you say in this related work section.
> > > > > Hope it helps.

---

> > > > > ### Author Response · Authors · 2019-11-09
> > > > > **Thank you for the suggestion**
> > > > >
> > > > > We really appreciate the detailed feedback. We will adjust the writing to incorporate your suggestions. We will do our best to improve the clarity of our presentation and let you know once the new revision is ready.

---

> > > > ### Author Response · Authors · 2019-11-09
> > > > **Weight Clipping**
> > > >
> > > > We are added additional ablation experiments comparing AWR with and without weight clipping. Results are available in the supplementary section D.1. To summarize, here are learning curves comparing the different policies:
> > > > https://imgur.com/a/FOsC0Cg
> > > > Policies trained without weight clipping are substantially more unstable, exhibiting drastic fluctuations in performance as a result of exploding gradients from excessively
> > > > large weights. Some training runs without clipping are terminated early due to exploding
> > > > gradients causing the networks to output NaNs. These experiments suggest that weight clipping is vital for ensuring stable training with AWR.

---

> ### Author Response · Authors · 2019-11-08
> **Initial response to R3**
>
> Re: Fitted Q-iteration by Advantage Weighted Regression
> Thank you for the pointer, We have revised the paper to include a discussion of “Fitted Q-iteration by Advantage Weighted Regression” [Neumann & Peters] at the end of Section 4 (highlighted in red). Neumann & Peters proposed a kernel-based fitted Q-iteration (FQI) algorithm. Though their method also uses exponentiated advantages as weights, their definition of the policy is different from our formulation:
> https://imgur.com/a/0DuRH7o
> The key difference is that in our method, the likelihood of an action is determined by both the likelihood of the sampling policy \mu and the exponentiated advantage, while the policy in Neumann & Peters depends only on the advantage. Therefore, the policy update from Neumann & Peters does not enforce a trust region penalty that ensures the new policy is similar to the sampling policy, which is crucial for obtaining a good estimate of the objective using off-policy data collected from the sampling policy. Our method is simpler, does not perform fitted Q-iteration, and incorporates experience for off-policy learning. Furthermore, we provide a principled derivation of the advantage weights from a conservative policy improvement perspective. We further extend this analysis to AWR with experience replay and demonstrate its effectiveness for batch RL, both of which were not presented in Neumann & Peters.
>
> Re: Self-Imitation
> SIL as described in https://arxiv.org/abs/1806.05635, augments policy gradient algorithms with an auxiliary behaviour cloning loss to reuse samples from past experiences. In contrast to SIL, AWR is a standalone algorithm, and does not need to be combined with an auxiliary RL algorithm. Also note that the weights in the self-imitation loss is given by max(adv, 0) rather than exp(adv).
>
> Re: PPO results
> We will update the PPO results with the data from John’s modified PPO implementation. We would like to point out that the original PPO results in our paper uses the standard implementation from OpenAI baselines, and the performance matches those reported in the original PPO paper and subsequent work that compares to PPO. As shown in our experiments, the difference between the deterministic and stochastic policies is fairly minor:
> https://imgur.com/a/mTOOZyc
> and most of the performance improvements are due to the other modifications (action squashing, reward scaling, etc…), which are not included in the standard PPO implementation:
> https://imgur.com/a/ny2rqFd

---

> > ### Comment · AnonReviewer3 · 2019-11-14
> > **Performance of TD3 in Fig. 3**
> >
> > A late comment (sorry) which is also minor: I have seen papers where TD3 reaches 12000 on HalfCheetah-v2. Any idea why yours does not perform well? Why did you stop this run earlier than the other ones? Given my doubts on Half-Cheetah, it also raises doubts about the performance of TD3 on Walker-2D, which seems to collapse.

---

> > > ### Author Response · Authors · 2019-11-14
> > > **TD3**
> > >
> > > We used the public implementation of TD3 available in RLKit: https://github.com/vitchyr/rlkit. There is a large variation in the performance of TD3 across different runs. As we have discussed in the other posts, methods that directly use a Q-function to update the policy are more susceptible to bias in the learned Q-function, which could be a factor in the performance of TD3 that we observe. Please note that the performance of TD3 does not collapse on Walker2D. In fact, it achieves a respectable score of 4212.
> > >
> > > Some of the TD3 runs were terminated earlier than others due to the slow wall-clock time of the TD3 code. We will update TD3 learning curves with longer runs once we have the time to train the policies for more time.
> > >
> > > We would like to emphasize again, that the goal of our experiments is not to show whether one algorithm is necessarily better than another. It is to show that a simple off-policy RL algorithm that uses supervised learning as subroutines can be competitive with current state-of-the-art techniques.

---

> ### Author Response · Authors · 2019-11-10
> **Neumann & Peters comparison**
>
> We have conducted additional experiments comparing AWR to the method from Neumann & Peters. Please refer to the general post for details:
> https://openreview.net/forum?id=H1gdF34FvS&noteId=H1xClC-Lsr

---

> ### Author Response · Authors · 2019-11-12
> **Updates to the submission**
>
> Thank you again for the detailed feedback. Here is a summary of the changes we have made to the latest draft of the submission:
>
> 1) We have performed additional experiments comparing our methods to FQI-by-AWR and results are available here https://openreview.net/forum?id=H1gdF34FvS&noteId=H1xClC-Lsr. These experiments will be added to the paper once we have more time to run additional random seeds.
>
> 2) A more thorough discussion of additional prior work has been added to Section 4.
>
> 3) We have expanded the preliminary section (Section 2) to include a more detailed review of RWR and EM algorithms, as well as a summary of their differences as compared to policy gradient algorithms.
>
> 4) We have added a discussion of the differences between our definition of the policy and the policy in FQI-by-AWR to Section 3.1.
>
> 5) We have included a more in-depth discussion of the differences between the AWR update and the standard policy gradient update to the supplementary material (Section D).
>
> 6) The additional ablation experiments for weight clipping that you requested for have been added to Section E.1 in the supplementary material.
>
> We hope these changes and experiments have helped to address your concerns. When you get a chance to, please update your review to reflect these changes, and let us know if you have any additional feedback.

---

> > ### Comment · AnonReviewer3 · 2019-11-12
> > **Equations mixed up**
> >
> > In the addition to p. 4, unless I got it wrong you repeated your \pi equation instead of stating Neumann's.
> >
> > Asking me to update my review is the AC's job, not yours. ;)
> > I'll do so after the discussion with the other reviewers, but I can already say that I appreciated your efforts in clarifying your paper.

---

> > > ### Author Response · Authors · 2019-11-12
> > > **Neumann & Peters equation**
> > >
> > > The equation in 3.1 for the definition of the policy from Neumann & Peters should be correct. Notice that the likelihood of an action depends only on its advantage and the likelihood under the sampling distribution is not present. Are you referring to some other mix-up?

---

> > > > ### Comment · AnonReviewer3 · 2019-11-12
> > > > **Neumann & Peters equation**
> > > >
> > > > Let me be more explicit about the mix-up.
> > > >
> > > > I'm refering to the sentence in red, page 4, which starts with "A similar advantage-weighting scheme has been previously used for fitted Q-iteration(Neumann & Peters, 2009), where the policy is given by *EQ*", where, if I understood correctly, *EQ* is supposed to be Neumann & Peters equation. But as a *EQ* I see exactly your Eq. (8) and not the Equation you were giving for Neumann & Peters equation there:
> > > > https://imgur.com/a/0DuRH7o
> > > >
> > > > Do you agree that there is a mix-up now?

---

> > > > ### Author Response · Authors · 2019-11-12
> > > > **The equation should be correct**
> > > >
> > > > Ok, seems like we are talking about the same equation. The equation in 3.1 should be correct. We adjusted the notation to be consistent with the notation we use in this paper, but it is saying the same thing as the equation in Neumann & Peters. For your convenience, we have included a side-by-side comparison of the three equations here:
> > > > https://imgur.com/a/oEAqBhP
> > > > Note that the new equation is different from our Equation 8. It doesn't not include the likelihood of an action under \mu. It is however semantically the same as equation 11 from Neumann & Peters, with only some notation differences. We replaced the normalization constant in the denominator with Z(s), the temperature was changed from \tau to \beta, and we also expanded the definition of the advantage A(s, a) = R_{s, a} - V(s).

---

> > > > > ### Comment · AnonReviewer3 · 2019-11-14
> > > > > **OK, no mix-up**
> > > > >
> > > > > Thank you, I'm now convinced, I have been too fast when looking at these equations.

---

### Official Review · AnonReviewer1 · 2019-10-23
**Official Blind Review #1**

**Rating:** 3

**Review:**

This paper proposes an off-policy reinforcement learning method in which the model parameters have been updated using the regression style loss function. Specifically, this method uses two regression update steps: one update value function and another one update policy using weighted regression. To compare the proposed method with others [main comprasion], 6 MuJoCo tasks are used for continuous control and LunarLander-v2 for discrete space.

-- Even though this paper has done a good job in terms of running different experiments, the selection of some of the benchmarks seems arbitrary. For example, for discrete action space, this paper uses LunarLander which is rarely used in any papers so it makes very difficult to draw a conclusion based on these results. Common 49 Atari-2600 games should have been used for comparison. The same thing about experiments in section 5.3 is true too as those tasks are not that well-known.

-- The proposed method doesn't outperform previous off-policy methods on Mujoco task (Table 1). Since the main claim of this paper is a new off-policy method, outperforming the previous off-policy methods is a fair game. The current results are not convincing enough.

-- There are significant overlaps between this paper, "Fitted Q-iteration by Advantage Weighted Regression", "Model-Free Preference-Based Reinforcement Learning ", and "Reinforcement learning by reward-weighted regression for operational space control" which makes the contribution of this paper very incremental.

-- The authors used only 5 seeds to run Mujoco experiments. Given the sensitivity of Mujoco for different starting points, the experiments should have been run at least with 10 different seeds.

Questions:
1) Shouldn't be an importance sampling ratio between \pi and \mu in the equations? starting from eq.5.
2) Does the algorithm optimize the respect to $w$ as well? (eq. 15) if yes, why it is not mentioned in algorithm 1? Plus, since $d(s)$ is uniform dist. (at least this is assumed for implementation), eq. 14,15,and 16 (wherever there is d(s)), those can be simplified, e.g. \hat{V} = \sum(w_i V_i), wouldn't be better just introduced simplified version rather than current ones? (referring to the only equation above section 3.3)
3) Is this the same code used to report results in this paper? if yes, I didn't see any seed assignment in the code?! and what is "action_std" in the code?

There are a couple of recent works in merging on-policy with off-policy updates which you might want to cite them.



**Experience Assessment:**

I have published one or two papers in this area.

**Review Assessment: Checking Correctness Of Derivations And Theory:**

I carefully checked the derivations and theory.

**Review Assessment: Checking Correctness Of Experiments:**

I assessed the sensibility of the experiments.

**Review Assessment: Thoroughness In Paper Reading:**

I read the paper thoroughly.

---

> ### Author Response · Authors · 2019-11-08
> **Initial response to R1**
>
> Initial response to R1
> Thank you for your feedback, we will aim to run the following additional experiments to address your questions:
> 1) We will perform experiments that directly compare AWR with REPS and FQI by AWR [Neumann & Peters].
> 2) We will include additional tasks with discrete action spaces such as Cartpole-v1, which is more commonly used.
> Please let us know if there are any additional experiments that you would find helpful.
>
> Re: Fitted Q-iteration by Advantage Weighted Regression
> Thank you for the pointer, We have revised the paper to include a discussion of “Fitted Q-iteration by Advantage Weighted Regression” [Neumann & Peters] in Section 4 (highlighted in red). Neumann & Peters proposed a kernel-based fitted Q-iteration algorithm. Though their method also uses exponentiated advantages as weights, their definition of the policy is different from ours:
> https://imgur.com/a/0DuRH7o
> The key difference is that in our method, the likelihood of an action is determined by both the likelihood of the sampling policy \mu and the exponentiated advantage, while the policy in Neumann & Peters depends only on the advantage. Therefore, the policy update from Neumann & Peters does not enforce a trust region penalty that ensures the new policy is similar to the sampling policy, which is crucial for obtaining a good estimate of the objective using off-policy data collected from the sampling policy. Our method is simpler, does not perform fitted Q-iteration, and incorporates experience for off-policy learning. Furthermore, we provide a principled derivation of the advantage weights from a conservative policy improvement perspective. We further extend this analysis to AWR with experience replay and demonstrate its effectiveness for batch RL, both of which were not presented in these prior works. "Model-Free Preference-Based Reinforcement Learning" uses REPS as its learning algorithm, and a discussion of REPS is available in Section 4. We will perform additional experiments to directly compare AWR with REPS and FQI by AWR.
>
> Re: Not outperforming previous methods
> While AWR does not outperform previous methods on all tasks, which we acknowledge in the paper, we would like to emphasize that our goal is to show that a simple off-policy method that uses supervised regression as subroutines can in fact be competitive with a number of current state-of-the-art algorithms. We believe that simple and effective RL algorithms are of interest to the ICLR community, and beating all state-of-the-art methods is not a prerequisite. We also demonstrate AWR’s effectiveness in the fully off-policy batch RL setting, where previous methods such DDPG and SAC perform poorly. Furthermore, AWR does not require the additional complexity of batch RL methods such as BCQ and BEAR.
>
> Re: Benchmarks
> We will also include additional tasks with discrete action spaces such as Cartpole-v1, which is a more commonly used benchmark. Note, our focus is primarily on continuous control tasks, and the addition of the discrete tasks is mainly to show that AWR can also be easily applied to discrete actions. The motivation for the tasks in section 5.3 is to show that AWR is also effective for controlling complex agents with larger numbers of degrees-of-freedom. Since the agents in standard benchmark tasks are relatively simple, we opt to use new environments to better demonstrate this capability.
>
> Re: additional random seeds
> We have ran additional experiments for AWR with 10 different seeds. Here are learning curves comparing the results with 5 seeds and 10 seeds. The results are similar. We will update the paper to include results with 10 seeds.
> https://imgur.com/a/yLntOzf
>
> Re: Q1 Importance sampling in Equation 5
> Importance sampling (IS) is not required in Equation 5. The objective itself does not require IS. For policy gradient methods like TRPO and PPO, IS is used to estimate the policy gradient from samples. In the AWR formulation, the solution of the Lagrangian in Equation 8 yields an update that does not require importance sampling, but it does require that \mu and \pi are similar, which is enforced by the trust region constraint (Equation 6).
>
> Re: Q2 Optimize w and uniform d(s)
> The algorithm does not optimize with respect to w. w_i represents the probability of selecting samples from a particular policy \pi_i from the replay buffer, which is a constant. d(s) is also not a uniform distribution over states, it represents the state-marginal of the sampling distribution. In our implementation, we sample uniformly from the replay buffer, which does not result in a uniform d(s), since some states might be visited more frequently than others.
>
> Re: code
> This is the same code used in the experiments. We do not manually assign random seeds to each run, instead we use python’s default random seed initialization, which assigns a different random seed to each execution of the problem. “action_std” specifies the standard deviation of the gaussian action distribution.

---

> ### Author Response · Authors · 2019-11-10
> **Neumann & Peters comparison**
>
> We have conducted additional experiments comparing AWR to the method from Neumann & Peters. Please refer to the general post for details:
> https://openreview.net/forum?id=H1gdF34FvS&noteId=H1xClC-Lsr

---

### Official Review · AnonReviewer2 · 2019-10-23
**Official Blind Review #2**

**Rating:** 6

**Review:**


# Summary
The paper shows that good old reward weighted regression (RWR) with value-function baseline is still state-of-the-art algorithm.

# Decision
The paper is well-written and provides many evaluations. The contribution should be articulated more carefully, though, taking into account that most algorithmic ideas are present in prior work (https://openreview.net/forum?id=H1gdF34FvS&noteId=Bkxi11nsdr). Perhaps, the experience replay part is somewhat novel. It seems emphasizing more that the aim is to show that simple methods are competitive rather than focusing on novelty could be a good idea.

Provided that the authors incorporate the feedback of the other reviewers and update the paper accordingly, it will make a good contribution.


**Experience Assessment:**

I have published one or two papers in this area.

**Review Assessment: Checking Correctness Of Derivations And Theory:**

I assessed the sensibility of the derivations and theory.

**Review Assessment: Checking Correctness Of Experiments:**

I assessed the sensibility of the experiments.

**Review Assessment: Thoroughness In Paper Reading:**

I made a quick assessment of this paper.

---

> ### Author Response · Authors · 2019-11-08
> **Initial response to R2**
>
> Thank you for the feedback, we will improve the writing to more clearly articulate the contribution of this work and include a more thorough discussion to contrast our method with related techniques. We have revised the paper to include a discussion of these prior works at the end of Section 4 (highlighted in red). We will also perform experiments that directly compare AWR with REPS and FQI by AWR [Neumann & Peters].
>
> While exponentiated-advantage weights have been used in a number of prior work, we present a principled derivation of AWR from a conservative policy improvement perspective, and also provide an analysis of AWR when combined with experience replay. We show that a number of simple design decisions, such as the use of TD-lambda and experience replay, enables AWR to achieve competitive performance with state-of-the-art algorithms both for RL and batch RL settings.

---

> ### Author Response · Authors · 2019-11-10
> **Additional experiments**
>
> We have conducted additional experiments comparing AWR to the method from Neumann & Peters. Please refer to the general post for details:
> https://openreview.net/forum?id=H1gdF34FvS&noteId=H1xClC-Lsr

---

### Public Comment · ~John_Schulman1 · 2019-10-07
**Nice paper, but PPO comparisons are flawed**

Hi authors,

I enjoyed this paper; I think it's the cleanest version of a cluster of ideas for weighted maximum likelihood objectives. (MAP-PO and self imitation learning are also in this cluster, and you might want to cite SIL.)

But the comparisons against PPO are not meaningful. You're comparing PPO's stochastic policy against the zero-noise deterministic policies from the other algorithms. By evaluating PPO in the same way, I get learning curves similar to your SAC learning curves: https://imgur.com/fTxLGWi (x axis units = 10^6 steps). Also, you're using different hyperparameters per environment, which seems improper.

Here's the code that produced this result, which is a minor modification of ppo1/run_mujoco.py in baselines: https://gist.github.com/joschu/852b04e985fe6bd74ed7557e83e6538a . (Also squashing the action space to enforce limits, as SAC implementations usually do, along with reward scaling.) If you want, I am happy to run the code myself and provide csv files for inclusion in your plots.

In addition, your humanoid learning curve shows PPO failing to learn anything. If you plan to include this comparison, I suggest that you run train_humanoid.py from https://github.com/openai/baselines/tree/master/baselines/ppo1 as described in the readme, and add the deterministic eval that I've provided in the gist.

I think the comparisons would be more meaningful if you used the same codebase for both algorithms but swapped only the loss functions (and separately optimized the hyperparams for each algorithm).

---

> ### Author Response · Authors · 2019-10-08
> **PPO Results**
>
> Hi John,
>
> Thank you for the feedback, and the code. We ran the code that you linked, but our results do not match the ones you posted. In fact, our original PPO comparison (which was included in the paper) performed substantially better. We provide learning curves for both versions here:
> https://imgur.com/a/UWamd6N
> "Original" refers to using the original baselines PPO code, and "Modified" refers to the script you provided. We have also asked other people run your script and they observe similar results to ours. We have ran your script without any modifications, but were unable to  reproduce the results you posted, and those results do not appear to match the figures reported in the original PPO paper. We are not aware of any other previously published papers or publicly available code that generate results similar to the ones that you posted.
>
> The PPO performance figures in our comparisons match those reported in the original PPO paper. We also find that  the performance of the stochastic and deterministic PPO policies from the baselines implementation are fairly similar:
> https://imgur.com/a/mTOOZyc
> However, we will update the performance statistics with those from the deterministic policies. The performance of the stochastic policy in your plots also appears to be substantially better than those from the baselines implementation. Perhaps there are other important differences besides using a deterministic vs stochastic policy?
>
> In the paper, we were using the ppo2 implementation from the baselines, since it seems that ppo1 might be deprecated:
> https://github.com/openai/baselines/issues/485
> The PPO results from the original paper more closely align with the ones reported in our comparisons.
>
> That said, we would like to emphasize that the purpose of our comparisons is to illustrate that AWR attains results that are comparable to current state-of-the-art methods. Our goal is not to show that a particular algorithm is necessarily better than another. It is well known that the particular details of the implementation can make a significant difference, especially on widely studied tasks like the gym benchmarks. So in the interest of reproducible, we prefer to use more standard and publicly available implementations of these algorithms.
>
> We would also like to point out that most hyperparameters are the same across the various environments. There are some parameters that varies a bit from environment to environment, such as the actor stepsizes for the humanoid and lunarlander, and the standard deviation of the action distribution.

---

> > ### Public Comment · ~John_Schulman1 · 2019-10-08
> > **followup**
> >
> > You need to multiply the return by 10 in your plot, because the RewScale wrapper divided it by 10. (Sorry, I should've mentioned that.) If you multiply by 10, the result looks like the plot I provided.
> >
> > On this task, the returns are poorly scaled, so the performance mostly depends on the details of normalization. I see that your code uses a separate stepsize for the policy and value function, with a much larger stepsize on a value function -- that seems like a way to achieve the same result.
> >
> > [EDIT] deleted previous comment because they appear in reverse order
> > [EDIT2] Modified the gist so now eval.csv gives the correctly scaled return.

---

> > > ### Author Response · Authors · 2019-10-15
> > > **new experiments**
> > >
> > > After scaling the returns, they seem to be similar to the plots you posted. We have conducted some ablation experiments on the modifications that were made to PPO. The overall conclusion appears to be that the difference between deterministic vs stochastic evaluation is fairly small for these tasks. Most of the performance improvements are a result of the other modifications. Here're some comparisons of the different modifications:
> > > https://imgur.com/a/ny2rqFd
> > >
> > > "Baselines": the original PPO code from OpenAI Baselines that we included in the paper (ppo2), which matches the performance in the original PPO paper.
> > > "Modified": uses the modified hyperparameters from your script (e.g. larger batch size, as well as using ppo1 instead of ppo2).
> > > "R Scale": applying reward scaling.
> > > "A Squash": applying action squashing.
> > >
> > > Reward scaling and action squashing seems to be responsible for most of the improvements. AWR still shows comparable performance on most of the tasks. We will update the paper to include these new results with the modified PPO implementation.

---

### Public Comment · ~Zhaoming_Xie1 · 2019-10-07
**TD3 humanoid comparison**

Just want to mention that in the original TD3 implementation, there is no normalization for the input to the network. That is probably why it performs so bad on the Humanoid task. The humanoid task include contact force to the state for some unknown reason and can have magnitude up to 1000. This is really bad for neural network without normalization. I think for fair comparison, similar normalization schemes should be employed for TD3.

---

> ### Author Response · Authors · 2019-10-08
> **TD3**
>
> Hi Zhaoming,
>
> Thank you for the suggestion and insight. We will look to try out input normalization for TD3. We would like to point out that it is generally standard practice to use publicly available code when comparing to prior algorithms. Since implementation details can have a significant effect on the performance of RL algorithms, in the interest of reproducibility, we prefer to use more standard and publicly available implementations of these algorithms in  our experiments. We would like to emphasize that the goal of our comparisons is to show that AWR can achieve comparable performance to current state-of-the-art methods, it is not to show whether one algorithm is necessarily better than another.

---

> > ### Public Comment · ~Zhaoming_Xie1 · 2019-10-08
> > **TD3**
> >
> > Thanks for the update. I think it will be beneficial to the community to compare each method in a fair way, so that people will have the insight as to which algorithm to use for their own problems.
> >
> > But I completely understand that this is out of the scope of this paper.

---

> > > ### Author Response · Authors · 2019-10-18
> > > **observation normalization**
> > >
> > > Thanks again for the suggestion. We have tried applying observation normalization to TD3 for the humanoid. Here's a performance comparison when training with and without normalization:
> > > https://imgur.com/a/ss7baug
> > > For this task, it doesn't seem to lead to a noticeable improvement in performance.

---

### Public Comment · ~Gerhard_Neumann1 · 2019-10-09
**A few missing references...**

Its a nice paper and the results look promising, but it should be put in a proper context. We introduced Advantage weighted regression already 10 years ago, although in a different algorithmic context (for fitted Q-iteration), see

https://papers.nips.cc/paper/3501-fitted-q-iteration-by-advantage-weighted-regression

I was a bit surprised that this paper is not cited. We also used already baselines in REPS like formulations, for example in the actor critic REPS algorithm

https://www.aaai.org/ocs/index.php/AAAI/AAAI16/paper/download/12247/11865

Again the setup is a bit different (its a kernel based algorithm and the baseline emerges from a constraint), but I think it would be fair to cite it and discuss the similarities.

---

> ### Author Response · Authors · 2019-10-10
> **references**
>
> Thank you for the references! We indeed missed this paper when we were preparing our submission, and we agree that this is a very relevant reference. We've added this reference to the paper, along with a discussion. We cannot update the openreview submission at this time, but will include an updated related work section in the final.
>
> We definitely agree that this prior paper also uses an advantage weighting formulation, though in contrast with this work, our paper does not perform fitted Q-iteration, instead opting for the simplest possible approach for policy search with off-policy data.

---

### Public Comment · ~John_Schulman1 · 2019-10-18
**looks good**

Great, thanks for putting in the time to do these thorough experiments!

---

### Public Comment · ~Jost_Tobias_Springenberg2 · 2019-11-06
**An interesting paper but perhaps even closer prior work is missing ?**

I finally had the time to read through this paper in detail and it seems to be a very nice empirical evaluation of one of the simplest ways to perform a form of reward weighted (or in this case advantage weighted) regression in combination with modern best practices for RL.
I applaud that the authors went this route and resisted the temptation to make the algorithm more complicated than necessary. Among the myriad of papers recently proposed this is a refreshingly simple and well written paper. The only part that would be great to clarify is that I suspect beta to be a quite hard to choose parameter if the reward scales become different (i.e. in benchmarks that are less homogeneous than the ones considered in the paper).

However, after reading the paper two times I am scratching my head about how this is different from MARWIL ( https://papers.nips.cc/paper/7866-exponentially-weighted-imitation-learning-for-batched-historical-data ) - which already used learned value functions for calculating the advantage and a replay mechanism. The only difference that I can see is that this paper not only considers the offline/imitation learning setting (where data is generated from some expert or behavior policy) but the full RL setting (which was hinted as a logical next application in the MARWIL paper). Perhaps I am missing something ?

---

> ### Author Response · Authors · 2019-11-07
> **MARWIL**
>
> re: tuning temperature
> The temperature does require some tuning, but more automated methods can be used, such as dual gradient descent. However, this strategy would require manually specifying an upperbound on the trust region constraint, which becomes another hyperparameter that requires tuning, which can also vary across different tasks. Therefore, we opt for a simpler strategy and used a fixed temperature.
>
> re: MARWIL
> Thank you for the pointer to MARWIL, indeed the method does appear similar to AWR, but their method was demonstrated only for imitation learning, while we include both RL and batch RL tasks, and compare our method to a number of state-of-the-art RL and batch RL algorithms. We also provide an analysis of AWR when combined with experience replay, where the sampling distribution is modeled by a trajectory-level mixture of rollouts collected from different policies, which was not presented in MARWIL. Though our design decisions such as TD-lambda and experience replay are fairly simple, we show that they are important components for an effective RL algorithm.  These consideration may not be as important when dealing only with imitation learning.

---

### Author Response · Authors · 2019-11-10
**Neumann & Peters  Comparison**

We have implemented the FQI-by-AWR algorithm as described in Algorithm 1 from Neumann & Peters (https://papers.nips.cc/paper/3501-fitted-q-iteration-by-advantage-weighted-regression.pdf), but instead of kernel-based function approximators, we used neural networks, like in our method, to ensure a fair comparison -- we believe that this likely makes the method stronger. A link to the code is available here:
https://drive.google.com/file/d/1vZdYSm84tdqMy1etJ7rtPGdgho3Zn1Jq/view?usp=sharing

Learning curves comparing the two methods are available here:
https://imgur.com/a/YjfMQS9
So far, our AWR algorithm performs substantially better than FQI-by-AWR on the tasks considered, despite also been much simpler. To the best of our knowledge, we are not aware of any prior work that shows the algorithm from Neumann & Peters to be effective for deep RL on this suite of tasks.

Here is a highlight of the differences between their methods and ours:
1) FQI-by-AWR performs fitted Q-iteration by learning both a Q-function and a value function. Our method requires only learning a value function.

2) FQI-by-AWR updates the policy by computing advantages using the learned Q-function and value function, whereas AWR simply uses monte-carlo returns (+ TD lambda) and a learned value function to update the policy. FQI-by-AWR’s use of the Q-function for policy updates could explain some of the instability observed in our experiments, since it is more susceptible to bias in the learned Q-function. This instability is commonly observed in other Q-learning based algorithms and often requires a myriad of stabilization techniques (https://arxiv.org/pdf/1710.02298.pdf, https://arxiv.org/pdf/1802.09477.pdf), none of which are needed for AWR.

3) In addition to fitting a Q-function and a value function, FQI-by-AWR also requires fitting estimators for the mean and standard deviation of the advantage at each state, which is not needed for our method.

4) FQI-by-AWR fits the value function using a td update with 1-step bootstrapping, while AWR uses either simple Monte Carlo returns or an off-the-shelf TD-lambda estimator.

We will continue tuning the parameters of our implementation of FQI-by-AWR and run the algorithm with more random seeds. The paper will be updated with the additional comparisons. We emphasize that we made a best-faith effort to implement this method, and were able to get it to achieve non-trivial performance. We hope the insights in our work for producing a simple but effective off-policy RL algorithm will be of interest to the community.

---

### Author Response · Authors · 2019-11-13
**Regarding novelty**

In regard to novelty, one thing we would like to note is that the combination of principled theoretical backing and solid empirical results that show a generally known recipe (in this case RWR), with a few careful design choices not present in prior work, can result in an effective deep RL algorithm is very much in line with widely recognized previously published papers. The same criticism in regard to novelty could have been applied to TRPO — which extended the theoretical backing of natural policy gradient and showed that it worked in the deep RL setting. Also PPO, which implementation wise is a small and simple modification on the very well known importance sampling policy gradient. The same for DDPG, which is a small but critical modification on NFQCA. By the novelty standard in the reviews, none of these prior papers — now viewed as landmark papers in deep RL — would have passed the bar. We believe that demonstrating, for the first time, that an RWR-like recipe produces a competitive deep RL algorithm with simple subroutines for value and policy fitting, together with theoretical motivation, is a contribution of similar scope in the context of prior work

---

### Author Response · Authors · 2019-11-15
**Author Response: Update Summary**

We thank the reviewers for their constructive feedback. We have revised the paper to include additional discussion of prior work and quantitative results. We summarize the main updates below:

1) comparisons with Neumann & Peters (https://imgur.com/a/YjfMQS9)

2) additional discussion of prior work in Section 4, including FQI by AWR, SIL, and MARWIL

3) a more detailed review RWR and EM methods in Section 2

4) additional ablation experiments in Appendix E

5) additional PPO experiments based on John Schulman’s updated PPO implementation (https://imgur.com/a/ny2rqFd)

6) discussion of differences between policy gradient and AWR update in Appendix D

7) experiments with 10 random seeds instead of 5 (https://imgur.com/a/yLntOzf)

---

### Decision · Program_Chairs · 2019-12-19

**Decision:**

Reject

**Comment:**

This paper caused a lot of discussions before and after the rebuttal. The concerns are related to the novelty of this paper, which seems to be relatively limited. Since we do not have a champion among positive reviewers, and the overall score is not high enough, I cannot recommend its acceptance at this stage.